# Asymptotically Exact Error Characterization of Offline Policy Evaluation with Misspecified Linear Models

**Kohei Miyaguchi**
IBM Research – Tokyo
`miyaguchi@ibm.com`

## Abstract

We consider the problem of offline policy evaluation (OPE) with Markov decision processes (MDPs), where the goal is to estimate the utility of given decision-making policies based on static datasets. Recently, theoretical understanding of OPE has been rapidly advanced under (approximate) realizability assumptions, i.e., where the environments of interest are well approximated with the given hypothetical models. On the other hand, the OPE under unrealizability has not been well understood as much as in the realizable setting despite its importance in real-world applications. To address this issue, we study the behavior of a simple existing OPE method called the linear direct method (DM) under the unrealizability. Consequently, we obtain an asymptotically exact characterization of the OPE error in a doubly robust form. Leveraging this result, we also establish the nonparametric consistency of the tile-coding estimators under quite mild assumptions.

## 1 Introduction

We consider the problem of offline data-driven decision optimization, wherein static records of previous interactions between decision makers and the environmental system of interest are given. The possible application areas include autonomous driving vehicles, natural-language dialogue systems, recommender systems, financial portfolio optimization and healthcare treatment optimization.

The framework of offline reinforcement learning (RL) is one of the promising approaches to this task (Levine et al., 2020). In the standard RL, the environment and the decision-making policy are respectively modeled as Markov decision processes (MDPs) $\mathcal{M}$ and conditional distributions of actions $\pi$ (Sutton and Barto, 2018), where each series of consecutive interactions between $\mathcal{M}$ and $\pi$ are abstracted as a stochastic sequence of state $s$, action $a$ and reward $r$, called an *episode*. The objective of the offline RL is then formalized as the maximization of the policy value $J(\pi)$, the expected value of the total reward obtained from a single episode, given a static dataset of previous interactions.

The crucial part of the problem is that the dataset is static; No additional interaction with the environment is allowed. This constraint poses several unique challenges to the problem. First, the policies we are optimizing, i.e., the target policies, cannot be run in the actual environment. Second, the policies used to generate the dataset, i.e., the behavior policies, are often unknown and may be totally different from the target policies. Consequently, it is even difficult to accurately estimate the value of target policies. This is problematic especially in consideration of real-life applications involving financial costs and healthcare risks.

To address the issue of policy value estimation, the problem of offline policy evaluation (OPE) have been extensively studied in the literature. A class of OPE algorithms are referred as the direct methods (DMs), in which some characteristics of $\mathcal{M}$ are assumed to be *realizable* under some

35th Conference on Neural Information Processing Systems (NeurIPS 2021).

Table 1: Comparison of the consistency conditions of linear DM. 'Yes' in the Consistency column implies linear DM solves *any* OPE instances satisfying the corresponding conditions. The definitions of the terminologies (such as *compatibility*) and mathematical symbols are given in Section 2 and 3.

| | Realizability | Exploration | Additional cond. | Consistency |
|---|---|---|---|---|
| Duan et al. (2020) | Bellman operator | concentrability | - | yes |
| Uehara et al. (2020) | $Q^\pi$ **and** $\nu/\mu$ | bounded $\nu/\mu$ | - | yes |
| Our result | $Q^\pi$ **or** $\nu/\mu$ | concentrability | compatibility | yes |
| Amortila et al. (2020) | $Q^\pi$ | concentrability | - | no |

hypothetical models and $J(\pi)$ is estimated via a direct estimation of such characteristics. For example, in the fitted Q-evaluation (FQE) algorithm (Le et al., 2019), the policy Q-function is assumed to be well-approximated with a parametric function class and the OPE is reduced to the estimation of its parameters.

DMs are known to be empirically effective (Fu et al., 2021) if such realizability assumptions are satisfied and, more importantly, *the converse is also true* (Voloshin et al., 2019). However, the theoretical understanding of DMs under unrealizability is still in its active development. For example, several authors have recently studied OPE or offline RL under relatively weak or approximate realizability assumptions (Jin et al., 2020; Xie and Jiang, 2020; Wang et al., 2020) and consequently proposing new algorithms.

In this paper, we approach the problem of unrealizability in the opposite direction; we start with an existing OPE method, study its behavior under complete unrealizability and seek for the possibility of regaining its consistency (i.e., asymptotically achieving zero errors). More specifically, we investigate the properties of a simple DM with linear function approximation, which is equivalent with a number of existing algorithms such as LSTD$Q$ (Lagoudakis and Parr, 2003), FQE (Le et al., 2019) with linear function regressors, the marginalized importance sampling estimator (Yin and Wang, 2020) and DualDICE (Nachum et al., 2019) in tabular settings.

In particular, we first characterize the exact asymptotic error of linear DM under as weak assumptions as possible. It turns out the error is governed by an inner product of two approximation residuals $\mathcal{R}_B$ and $\mathcal{R}_\chi$,

$$\hat{J}(\pi) - J(\pi) \propto \mathbb{E}\left[\mathcal{R}_B(s,a)\mathcal{R}_\chi(s,a)\right] + \mathcal{O}\left(1/\sqrt{n}\right), \qquad \text{(informal)}$$

where they are corresponding to the unrealizable components of *the value function $Q^\pi$* and *the marginal density ratio $\nu/\mu$*, respectively. To the best of our knowledge, this is the first to show linear DM is doubly robust against model misspecification, i.e., consistent if either $\mathcal{R}_B = 0$ or $\mathcal{R}_\chi = 0$ hold (Table 1). Leveraging the above result, we also show that a linear DM with the tile-coding function approximation (Section 8.3.2, Sutton and Barto (2018)) is consistent under surprisingly mild conditions with appropriate tile-size scheduling.

The rest of the paper is organized as follows. In Section 2, we formalize the problem setting as well as the definition of the linear direct estimators. In Section 3, we present the main results, i.e., the asymptotic error analysis of the linear direct estimators and a construction of consistent nonparametric estimators as its application. In Section 4, we discuss related works with comparison to our results. Finally, in Section 5, we present concluding remarks, limitations and future directions. All the proofs of the propositions and the theorem are relegated to the appendix. See Section F for the proofs of the propositions. For the theorem, we present a proof sketch and the pointer to the full proof.

## 2 Preliminary

In Section 2.1, some notational conventions are introduced. The problem of OPE is then formalized in Section 2.2. Then, Section 2.3, 2.4 and 2.5 respectively introduce assumptions and definitions on the data-collecting processes, the environmental models and the class of estimators we will examine.

## 2.1 Basic Notation

We implicitly assume the spaces we encounter in this paper, such as the state space $\mathcal{S}$ and the action space $\mathcal{A}$, are equipped with respective metrics and base measures, each of which is a compact subset of either a Euclidean space with the Lebesgue measure, a discrete space with the counting measure or a product of those. We denote by $\int_{\mathcal{X}} f(x) \, dx$ the integration of function $f$ with respect to the base measure of $\mathcal{X}$. The subscript $\mathcal{X}$ may be omitted if it is obvious from the context. This way we can immediately generalize our results to both continuous and discrete spaces. Also, we denote the expectation of function $f$ with respect to probability density $p$ by $\langle f \rangle_p := \mathbb{E}_{x \sim p}[f(x)]$.

Let $[m] := \{1, 2, \ldots, m\}$ denote the set of integers from $1$ to $m$. Let $\|\cdot\|_p$ denote the $\ell^p$-norm for vectors and $\|A\|_{p \to q} := \sup_{x \neq 0} \|Ax\|_q / \|x\|_p$ the operator norms for matrices, with the convention $\|A\|_p := \|A\|_{p \to p}$, for all $1 \leq p, q \leq \infty$.

## 2.2 Problem Setup

The goal of OPE is to estimate the *value* of decision-making strategy based on a static dataset of interactions with the environment of interest, without directly knowing its mechanism.

The environment is modeled as a Markov decision process (MDP) $\mathcal{M} \equiv (\mathcal{S}, \mathcal{A}, p_0, p_T, p_r)$, where $\mathcal{S}$ is the state space, $\mathcal{A}$ the action space, $p_0(s)$ the initial state probability, $p_T(s'|s, a)$ the transition probability and $p_r(r|s, a)$ the $[0, 1]$-valued reward probability density function for $s, s' \in \mathcal{S}$, $a \in \mathcal{A}$, $r \in [0, 1]$. Here we assume $p_T$ and $p_r$ are unknown. On the other hand, the decision-making strategy is modeled as a *policy*, a state-conditional action distribution $\pi(a|s)$ for $s \in \mathcal{S}$, $a \in \mathcal{A}$.

The value of $\pi$ is measured with the expected cumulative reward

$$J(\pi) := \sum_{h=0}^{\infty} \gamma^h \left\langle P^h \bar{r} \right\rangle_{p_0^\pi}, \tag{1}$$

where $\gamma \in [0, 1)$ is the discounting factor, $P$ is the state-transition operator such that $(Pf)(s, a) = \int f(s', a') p_T(s'|s, a) \pi(a'|s') \, ds' \, da'$, $\bar{r}(s, a) := \int r \, p_r(r|s, a) \, dr$ is the expected reward function, and $p_0^\pi(s, a) := p_0(s) \pi(a|s)$ is the initial state-action distribution. In particular, $\langle P^h \bar{r} \rangle_{p_0^\pi}$ denotes the expected reward after $h$ transitions starting from $p_0^\pi$.

The policy value $J(\pi)$ is estimated based on a collection of transition records $\xi^n \equiv (\xi_1, ..., \xi_n) \in \mathcal{D}^n$ called an *offline* dataset, where $\mathcal{D} := \mathcal{S} \times \mathcal{A} \times [0, 1] \times \mathcal{S}$ is the space of transition records and $\xi_i \equiv (s_i, a_i, r_i, s_i') \in \mathcal{D}$, $i \in [n]$, is a transition record made of a preceding state-action pair $(s_i, a_i)$, the associated reward $r_i$, and the state after transition $s_i'$. The dataset $\xi^n$ is assumed to be an instantiation of the random variables $\Xi^n \equiv (\Xi_1, ..., \Xi_n)$, $\Xi_i \equiv (S_i, A_i, R_i, S_i')$, collected with interactions between the environment $\mathcal{M}$ and a *query distribution* $p_{\text{query}} \equiv \{p_{\text{query}(i)}(s, a|\xi^{i-1})\}_{i \in [n]}$ such that its distribution is given in a conditional fashion,

$$p(\xi_i|\xi^{i-1}) = p_{\text{query}(i)}(s_i, a_i|\xi^{i-1}) \, p_r(r_i|s_i, a_i) \, p_T(s_i'|s_i, a_i), \quad i \in [n].$$

Note that the notion of query distribution is so flexible that it admits $\xi^n$ to be a union of episodes generated with multiple nonstationary policies and even adversaries on the choice of state-action pairs.

**Definition 1** (OPE problem). *An instance of the offline policy evaluation problem is specified with $\mathcal{P}_{\text{OPE}} \equiv (\mathcal{M}, \pi, \gamma, p_{\text{query}})$, where the goal is to estimate the policy value $J(\pi)$ determined by $(\mathcal{M}, \pi, \gamma)$, given the input data $\xi^n$ generated with $(\mathcal{M}, p_{\text{query}})$, without knowing any of $p_T$, $p_r$ or $p_{\text{query}}$.*

## 2.3 Assumptions on Data-Collecting Process

To ensure the existence of reasonable estimators for $\mathcal{P}_{\text{OPE}}$, we pose a condition on the mixing of data-collecting process, i.e., conditions on $p_{\text{query}}$. We assume the amounts of mutual dependencies induced by $p_{\text{query}}$ between time-distant transition records are bounded.

**Assumption 1** ($G^*$-mixing dataset). *There exists a constant $G^* < \infty$ such that $\Xi^n$ is '$\phi$'-strong mixing with the coefficient $g(h)$ satisfying $1 + 2 \sum_{h=1}^{n} \sqrt{g(h)} \leq G^*$.*[1]

---

[1] The symbol '$\phi$' in the '$\phi$'-strong mixing has its root in statistics and completely unrelated to the feature mapping. We denote the mixing coefficient by $g(h)$ to avoid confusion with the feature mappings.

See Definition 17 (in the appendix) for the definition of the '$\phi$'-strong mixing coefficients. Typical examples satisfying Assumption 1 include datasets consisting of multiple short episodes and mixing Markov chains induced by stationary behavior policies.

**Proposition 1.** *The following statements are true.*

1. *Assume $\Xi^n$ consists of multiple independently collected episodes with length bounded by $H$, ordered in a consecutive manner. Then we have $G^* \leq 2H - 1$.*

2. *Let $p_{\mathrm{query}(i)}(s, a | \xi^{i-1}) = p_T(s|s_{i-1}, a_{i-1})\pi_b(a|s)$, $1 \leq i \leq n$, for some stationary behavior policy $\pi_b(a|s)$ and assume the resulting Markov chain has a finite mixing time $t_{mix} < \infty$. Then we have $G^* \leq 1 + 7t_{mix}$.*

Note that the definition of $G^*$-mixing is designed to be more general than these examples. In particular, it is more suitable for our query-distribution framework, which admits adversaries behind the choice of $(s_i, a_i)$-s or dynamically changing behavior policies.

Under Assumption 1, most properties of $p_{\mathrm{query}}$ is characterized with the marginal data density.

**Definition 2** (Marginal data density)**.** *Let $\mu(s, a)$ be the marginal data density, given by $\mu(s, a) := \frac{1}{n}\sum_{i=1}^{n}\mathbb{E}[p_{\mathrm{query}(i)}(s, a \,|\, \Xi^{i-1})]$ for $s \in \mathcal{S}$ and $a \in \mathcal{A}$.*

The marginal data density $\mu$ quantifies the expected frequency of visitation at each point $(s, a) \in \mathcal{S} \times \mathcal{A}$ made by the querying process. Thus, roughly speaking, $\mu(s, a)$ indicates that how likely the point $(s, a)$ will be sampled in $\Xi^n$.

## 2.4 (Possibly Misspecified) Environmental Model: Linear MDPs

We introduce linear MDPs, a simple class of the environmental models denoted by $\mathcal{H}_\phi$.[2] We also define the projection of $\mathcal{M}$ onto $\mathcal{H}_\phi$ as we are concerned with the unrealizable case, $\mathcal{M} \notin \mathcal{H}_\phi$.

A linear MDP $\mathcal{H}_\phi$ is formally defined via a bounded vector-valued function on the state-action space called a *feature mapping*, denoted by $\phi : \mathcal{S} \times \mathcal{A} \to \mathbb{R}^K$, $K \geq 1$. We assume the boundedness and the concentrability of the mapping as follows.

**Assumption 2** (Boundedness)**.** $\sup_{s \in \mathcal{S}, a \in \mathcal{A}} \|\phi(s, a)\|_2 \leq 1$.

**Assumption 3** (Concentrability)**.** *Let $\Sigma := \langle \phi\phi^\top \rangle_\mu$ be the feature covariance matrix and $c_* := \lambda_K(\Sigma)$ be its smallest eigenvalue. Then, $c_* > 0$.*

Note that the concentrability is a standard assumption ensuring all the dimensions of the feature space will be explored in the data-collecting process. A typical example satisfying the above assumptions is the tabular features.

**Remark 1.** $\phi$ *is said to be tabular if there exits a $K$-partition of $\mathcal{S} \times \mathcal{A}$, $\{\mathcal{P}_k\}_{k \in [K]}$, such that*

$$\phi_k(s, a) = \mathbb{I}\{(s, a) \in \mathcal{P}_k\}, \ k \in [K], \ s \in \mathcal{S}, \ a \in \mathcal{A},$$

*where $\mathbb{I}\{\cdot\}$ denotes the indicator function. The tabular features always satisfy Assumption 2. Moreover, it satisfies Assumption 3 if every cell is covered with the data marginal, $\min_{k \in [K]} \mathbb{P}_\mu(\mathcal{P}_k) > 0$, where $\mathbb{P}_\mu$ denotes the probability measure induced by $\mu$.*

Now, the class of $\phi$-linear MDPs is defined as follows.

**Definition 3** ($\phi$-linear MDPs)**.** *We say $\mathcal{M}$ is $\phi$-linear with respect to $\pi$,[3] if there exist $b \in \mathbb{R}^K$ and $F \in \mathbb{R}^{K \times K}$ such that*

$$\bar{r}(s, a) = b^\top \phi(s, a), \qquad\qquad (P\phi)(s, a) = F\phi(s, a) \qquad\qquad (2)$$

*for almost every $s \in \mathcal{S}$ and $a \in \mathcal{A}$. We refer to the set of all the $\phi$-linear MDPs as $\mathcal{H}_\phi$.*

---

[2]We never assume $\mathcal{H}_\phi$ contains the true environmental model. It is rather used to facilitate the construction of OPE estimators in Section 2.5.

[3]The $\phi$-linearity is the property of the MDP $\mathcal{M}$ *and* the policy $\pi$ since the transition operator $P$ depends on $\pi$. However, we omit the dependency on $\pi$ for brevity.

In other words, $\mathcal{M}$ is $\phi$-linear if both the reward distribution and the transition dynamics are linearly predictable in expectation with respect to $\phi$. This definition is motivated by the following proposition; if $\mathcal{M}$ is realizable as a member of $\mathcal{H}_\phi$, the problem of OPE is reduced to the estimation of $b$ and $F$.

**Proposition 2.** *Under Assumption 2 and 3, if $\mathcal{M} \in \mathcal{H}_\phi$, we have*

$$J(\pi) = b^\top (I - \gamma F)^{-1} x_0, \tag{3}$$

*where $x_0 := \int \phi(s, a) \, p_0^\pi(s, a) \, \mathrm{d}s \, \mathrm{d}a$.*

However, it is impractical to assume we know the mapping $\phi$ that attains the realizability with the environment of interest $\mathcal{M}$. Thus we introduce the projection of $\mathcal{M}$ onto $\mathcal{H}_\phi$.

**Definition 4** (Projection of MDP). *Let $D^2(b, F)$ be the parameter discrepancy of the $\phi$-linearity, given by*

$$D^2(b, F) := \mathbb{E}_{(s,a) \sim \mu} \left[ \left| \bar{r}(s, a) - b^\top \phi(s, a) \right|^2 + |(P\phi)(s, a) - F\phi(s, a)|^2 \right]. \tag{4}$$

*We refer to its minimizer as the projections of $\mathcal{M}$ onto $\mathcal{H}_\phi$, denoted by $(b^\sharp, F^\sharp)$.*

Note that the projection coincides with the true parameter $(b, F)$ if $\mathcal{M}$ is realizable. Throughout the paper, however, we consider the general case in which the true parameter may not exist, but $(b^\sharp, F^\sharp)$ always does.

## 2.5 Linear Direct Estimators

We finally introduce the linear direct estimator. The idea of the linear direct estimator is twofold. First, we approximately solve the minimization of (4) based on the sample $\xi^n$ to obtain the estimate of the projection, $(\hat{b}, \hat{F}) \approx (b^\sharp, F^\sharp)$. Then, we plug the estimate into (3) to get a policy value estimate, which seems reasonable if $\mathcal{M}$ is (approximately) realizable.

More precisely, the first step is formalized via the least squares method.

**Definition 5** (Empirical projection of MDP). *The empirical projection $(\hat{b}, \hat{F})$ is defined as the minimizer of the following cost function,[4]*

$$\mathcal{C}(b, F; \xi^n) := \frac{1}{n} \sum_{i=1}^n \left[ \left| r_i - b^\top \phi(s_i, a_i) \right|^2 + |\psi_\pi(s_i') - F\phi(s_i, a_i)|^2 \right].$$

*Here, $\psi_\pi(s) := \int \phi(s, a)\pi(a|s) \, \mathrm{d}a$ is the state-marginal feature mapping.*

This definition is justified as follows.

**Proposition 3.** *For all $b \in \mathbb{R}^K$ and $F \in \mathbb{R}^{K \times K}$, $\nabla_{b,F}\mathbb{E}[\mathcal{C}(b, F; \Xi^n)] = \nabla_{b,F} D^2(b, F)$.*

In other words, the gradient of the cost function coincides with that of the parameter discrepancy function in expectation and thus one can expect $(\hat{b}, \hat{F}) \to (b^\sharp, F^\sharp)$ in the large sample limit.

We have a closed form of the empirical projection.

**Proposition 4.** *Let $(\Phi, \Psi_\pi, \hat{r})$ be given by*

$$\Phi := [\phi(s_1, a_1), ..., \phi(s_n, a_n)]^\top, \qquad \Psi_\pi := [\psi_\pi(s_1'), ..., \psi_\pi(s_n')]^\top, \qquad \hat{r} := [r_1, ..., r_n]^\top.$$

*Then, the empirical projection is given by*

$$\hat{b} = \frac{1}{n}\hat{\Sigma}^+ \Phi^\top \hat{r}, \qquad\qquad \hat{F} = \frac{1}{n}\Psi_\pi^\top \Phi \hat{\Sigma}^+, \tag{5}$$

*where $\hat{\Sigma} := \frac{1}{n}\Phi^\top \Phi$ is the empirical covariance matrix and $\hat{\Sigma}^+$ is its pseudo-inverse.*

---

[4]If the sample size $n$ is small, the minimizer may not be unique. In this case, we admit multiple empirical projections.

---

**Algorithm 1** Linear Direct OPE

**Input:** Initial distribution $p_0$, target policy $\pi$, data $\xi^n$, feature mapping $\phi$
**Output:** Policy value estimate $\hat{J}(\pi)$
1: Compute the empirical projection $(\hat{b}, \hat{F})$ according to Proposition 4.
2: Compute $\hat{J}(\pi) = \hat{b}^\top (I - \gamma\hat{F})^{-1} x_0$, where $x_0 := \int \psi_\pi(s) p_0(s)\, \mathrm{d}s$.

---

**Algorithm 2** Linear Fitted Q-Evaluation

**Input:** Initial distribution $p_0$, target policy $\pi$, data $\xi^n$, feature mapping $\phi$, iteration number $H$
**Output:** Policy value estimate $\hat{J}_H(\pi)$
1: Let $\theta_0 := 0 \in \mathbb{R}^K$.
2: **for** $h = 1, 2, ..., H$ **do**
3:     Find $\theta_h$, the least-norm minimizer of $\frac{1}{n} \sum_{i=1}^{n} \left| r_i + \gamma\theta_{h-1}^\top \psi_\pi(s_i') - \theta_h^\top \phi(s_i, a_i) \right|^2$.
4: **end for**
5: Compute $\hat{J}_H(\pi) = \theta_H^\top x_0$, where $x_0 := \int \psi_\pi(s) p_0(s)\, \mathrm{d}s$.

---

The whole procedure is summarized in Algorithm 1. Note that $\psi_\pi(s)$ in $\Psi_\pi$ and $x_0$ is not necessarily tractable in a closed form. One can always resort to Monte-Carlo estimates $\psi_\pi(s) \approx \frac{1}{n_\psi} \sum_{\ell=1}^{n_\psi} \phi(s, a_\ell)$, where $a_\ell \sim \pi(a|s)$, $\ell \in [n_\psi]$, are i.i.d. samples. Proposition 3 still holds under this approximation. Also note that $\hat{J}(\pi)$ is undefined if $I - \gamma\hat{F}$ is singular. To avoid the undefined behavior and the numerical instability due to near-singularity, we assume $\phi$ is *compatible* with $\mathcal{P}_{\mathrm{OPE}}$ in the following sense.

**Assumption 4** (Compatibility). $F_\gamma^\sharp := (I - \gamma F^\sharp)^{-1}$ *exists.*

Note that the compatibility implies the well-definedness of $\hat{J}(\pi)$ with high probability for sufficiently large $n$ since $F^\sharp = \lim_{n\to\infty} \hat{F}$. We also discuss the interpretation, sufficient conditions and a statistical test of the compatibility in Section 3.1.1 and 3.1.2.

Algorithm 1 is equivalent to the LSTD$Q$ (Lagoudakis and Parr, 2003) algorithm and a number of equivalence relationships to recent OPE estimators are drawn in Duan et al. (2020). For the completeness, we show Algorithm 1 is equivalent to the limit of Fitted Q-Evaluation (Le et al., 2019) with linear function approximators, shown in Algorithm 2.

**Proposition 5.** *The output of Algorithm 1 $\hat{J}(\pi)$ is identical to the limit of that of Algorithm 2, $\lim_{H\to\infty} \hat{J}_H(\pi)$, if both exist.*

## 3 Main Results

First, we give an asymptotic characterization of the error $\hat{J}(\pi) - J(\pi)$, which sheds light on the doubly robust nature of linear DM. Second, leveraging the first result, we show novel consistency properties of a simple tile-coding estimator.

### 3.1 Asymptotic Error of Linear DM

As will be shown later, the dominant term of the OPE error is written as an inner product of two functions, namely the $\chi$-residual function and the Bellman residual function. To introduce these residual functions, we begin with the definitions of the $\phi$-spanned function space, the $\phi$-semi norm and the marginal target density.

**Definition 6** ($\phi$-spanned function space). *We denote by $\mathcal{F}_\phi$ the normed function space spanned by $\phi_1, ..., \phi_K$, i.e., $\mathcal{F}_\phi := \left\{ (\theta^\top \phi) : \mathcal{S} \times \mathcal{A} \to \mathbb{R} \mid \theta \in \mathbb{R}^K \right\}$.*

**Definition 7** ($\phi$-semi norm). *For any real-valued functions over the state-action space $f : \mathcal{S} \times \mathcal{A} \to \mathbb{R}$, the $\phi$-semi norm of $f$ is given by $|f|_\phi := \| \langle \phi f \rangle_\mu \|_2$.*

**Definition 8** (Marginal target density). *Let $\nu(s, a)$ be the marginal target density, given by $\nu(s, a) := (1 - \gamma) \sum_{h=0}^{\infty} \gamma^h (P^{\dagger h} p_0^\pi)(s, a)$, where $P^\dagger$ is the adjoint operator of $P$.*

Note that $(P^{\dagger h} p_0^\pi)(s, a)$ denotes the state-action density after $h$ transitions starting from $p_0$. Thus, $\nu$ can be thought of as the relative frequency of the state-action visitations in the target episode with horizon-dependent multiplicative weights $\gamma^h$.

Then, two residual functions are defined as follows.

**Definition 9** ($\chi$-residual function)**.** *The $\chi$-residual function is defined as*

$$\mathcal{R}_\chi(s, a) := \frac{\nu(s, a)}{\mu(s, a)} - w^\sharp(s, a),$$

*where $w^\sharp$ is the minimizer of $\mathcal{L}_\chi(w) := \langle (\frac{\nu}{\mu} - w)^2 \rangle_\mu$ in $\mathcal{F}_\phi$.*

**Definition 10** (Bellman residual function)**.** *The Bellman residual function is defined as*

$$\mathcal{R}_B(s, a) := (BQ^\sharp)(s, a) - Q^\sharp(s, a),$$

*where $B : Q \mapsto \bar{r} + \gamma P Q$ is the Bellman operator and $Q^\sharp$ is the minimizer of $\mathcal{L}_B(Q) := |BQ - Q|_\phi^2$ in $\mathcal{F}_\phi$.*

Note that these functions are 'residual' since they are the remainders of the projection of some functions onto $\mathcal{F}_\phi$. $\mathcal{R}_\chi$ is the residual of the density ratio $\nu/\mu$ and $\mathcal{R}_B$ is the residual of the Bellman equation. Also note that the projection of the Bellman equation is *coarse-grained* as $|f|_\phi = 0$ does not necessarily imply $f = 0$ (therefore it is a *semi*-norm). We will discuss further interpretation of these residual functions in Section 3.1.1.

Now we are ready to state our first result.

**Theorem 6.** *Suppose Assumption 1, 2, 3 and 4 hold. Then, we have the almost-sure convergence*

$$\hat{J}(\pi) - J(\pi) \overset{n \to \infty}{\longrightarrow} -\frac{1}{1 - \gamma} \langle \mathcal{R}_B \mathcal{R}_\chi \rangle_\mu \tag{6}$$

*in a $\mathrm{poly}(\frac{1}{1-\gamma}, \frac{1}{n}, G^*, \frac{1}{c_*}, \|F_\gamma^\sharp\|_2)$ rate, uniformly with respect to the choice of $p_0$. Moreover, the convergence is also uniform with respect to the choice of $\pi$, if $\sup_\pi \|F_\gamma^\sharp\|_2 < \infty$.*

*Proof.* (Sketch.) It is directly derived from the non-asymptotic bound (Theorem 13, in the appendix), whose proof strategy is to decompose the error by $\hat{J}(\pi) - J(\pi) = (J^\sharp(\pi) - J(\pi)) + (\hat{J}(\pi) - J^\sharp(\pi))$, where $J^\sharp(\pi) := \langle Q^\sharp \rangle_{p_0^\pi}$ is the projected policy value, and evaluate these terms separately. The limit (6) is obtained by evaluating the first term and the second term vanishes in a rate of $\mathcal{O}(1/\sqrt{n})$. In particular, the key step in the evaluation of the first term is the following series of identities,

$$J^\sharp(\pi) - J(\pi) = \ldots = -\langle \mathcal{R}_B \rangle_\nu = -\left\langle \mathcal{R}_B \frac{\nu}{\mu} \right\rangle_\mu = -\left\langle \mathcal{R}_B \left(\frac{\nu}{\mu} - f\right) \right\rangle_\mu, \quad \forall f \in \mathcal{F}_\phi,$$

where the last identity is what allows us to fit arbitrary function in $\mathcal{F}_\phi$ away from $\nu/\mu$, which is made possible with $\mathcal{R}_B$ being in the orthogonal complement of $\mathcal{F}_\phi$. The full proof is deferred to Section A. $\qquad\square$

### 3.1.1 Implications and Interpretations

**Consistency under Q-function realizability (and necessity of the compatibility).** Theorem 6 shows the necessary and sufficient condition for the consistency of compatible linear DM. Specifically, $J(\pi)$ is consistent if either $\mathcal{R}_B = 0$ or $\mathcal{R}_\chi = 0$. The first condition $\mathcal{R}_B = 0$ can be interpreted as the so-called Q-function realizability.

**Proposition 7.** *Let $Q^\pi := \sum_{h \geq 0} \gamma^h \langle P^h \bar{r} \rangle_{p_0^\pi}$ be the policy Q-function. Then, under Assumption 2, 3 and 4, we have $\mathcal{R}_B = 0$ if and only if $Q^\pi \in \mathcal{F}_\phi$.*

This is interesting since $\mathcal{R}_B$ preserves the full information of $Q^\pi$, the unique solution to the Bellman equation, even though it is the result of the coarse-grained projection (Definition 10). Moreover, compared with the existing results on the hardness of OPE under realizability (Wang et al., 2020; Amortila et al., 2020), Proposition 7 suggests the compatibility is the key for the consistency of linear DM. In particular, Amortila et al. (2020) showed there exists a class of hard instances where consistent OPE is impossible if we just assume the Q-function realizability and the concentrability. On the other hand, Theorem 6 and Proposition 7 show we can eliminate such instances if we additionally assume the compatibility (Table 1).

**Consistency under density-ratio realizability.** Theorem 6 implies another route to achieve the consistency, namely $\mathcal{R}_\chi = 0$. By definition, it is trivial that $\mathcal{R}_\chi = 0$ is equivalent to $\nu/\mu \in \mathcal{F}_\phi$.

**Proposition 8.** *Under Assumption 2 and 3, we have $\mathcal{R}_\chi = 0$ if and only if $\nu/\mu \in \mathcal{F}_\phi$.*

It suggests we have a consistent estimate of $J(\pi)$ when the density ratio $\nu/\mu$ is realizable even if the true value function $Q^\pi$ is not at all realizable. This is interesting since linear DM is equivalent to linear FQE, which is designed to estimate value function $Q^\pi$. In another words, linear FQE is *doubly robust* in bias without estimating the density ratio. See Section 4 for further comparison with previous double robustness results. Moreover, the density-ratio realizability behaves better than the Q-function realizability. In particular, if we have a non-decreasing sequence of linear function classes $\{\mathcal{F}_{\phi^{(1)}} \subset \mathcal{F}_{\phi^{(2)}} \subset \cdots\}$, the corresponding sequence of $\chi$-residuals $\{\mathcal{R}_\chi^{(1)}, \mathcal{R}_\chi^{(2)}, \cdots\}$ is also non-increasing, i.e., $\|\mathcal{R}_\chi^{(m)}\|_{L^2(\mu)}$ is the non-increasing function of $m \geq 1$. This does not hold in general with the Bellman residual $\mathcal{R}_B$ because the projection metric depends on $\phi$.

**Convergence without concentrability.** A convergence result similar to Theorem 6 can be established without the concentrability condition (Assumption 3), i.e., even if the covariance matrix $\Sigma$ is singular. In this case, the projections $b^\sharp$, $F^\sharp$, $w^\sharp$ and $Q^\sharp$ is modified to the *least-norm* minimizers and the rate of convergence depends on the minimum *nonzero* eigenvalue of $\Sigma$. See Definition 13 and Theorem 13 in the appendix for the detailed statement.

**Linear DM is $\mathcal{L}_B(Q)$-minimization.** As illustrated in the sketch, the proof of Theorem 6 gives the large sample limit of $\hat{J}(\pi)$, namely $\hat{J}(\pi) \to J^\sharp(\pi) = \langle Q^\sharp \rangle_{p_0^\pi}$. Therefore, since $Q^\sharp$ is a minimizer of $\mathcal{L}_B(Q)$ (Definition 10), linear DM is asymptotically equivalent to solving $\min_{Q \in \mathcal{F}_\phi} \mathcal{L}_B(Q)$. This observation is closely related to the kernel-loss interpretation of the temporal difference methods (Corollary 3.5 in Feng et al. (2019)).

### 3.1.2 Sufficient Conditions for Compatibility and Uniform Boundedness of $F_\gamma^\sharp$

In a practical sense, the compatibility is a necessary (not just sufficient) condition for the asymptotic convergence of linear DM; if $I - \gamma F^\sharp$ is singular, the smallest singular value of $I - \gamma \hat{F}$ approaches to zero and hence its inverse $(I - \gamma \hat{F})^{-1}$ is divergent and unstable (if not nonexistent) as well as the estimate $\hat{J}(\pi)$. This is seen from the concentration $\lim_{n \to \infty} \hat{F} = F^\sharp$ (see Proposition 19 in the appendix) and the continuity of singular values.

However, it is difficult to *completely* characterize when $F_\gamma^\sharp$ exists or is uniformly bounded. Below, we show a sufficient condition for the uniform boundedness which depends only on $\mu$ and $\Sigma$.

**Proposition 9.** *Under Assumption 2 and 3, we have $\sup_\pi \|F_\gamma^\sharp\|_2 < \infty$ if*

1. *There exists $v_0 \in \mathbb{R}^K$ such that $v_0^\top \phi(\cdot, \cdot) \equiv 1$, and*

2. *$\phi(s, a) \Sigma^{-1} \phi(s', a') \geq 0$ for all $(s, a), (s', a') \in \mathcal{S} \times \mathcal{A}$.*

Note that both assumptions hold with the tabular features $\phi$, i.e., *tabular features are always compatible*. In general, the first condition can be satisfied by expanding $\phi$ with an extra 'bias' dimension whose value is always one. On the other hand, the second condition is dependent on $\phi$ and $\Sigma$ in a nontrivial manner.

Even if it is difficult to analytically guarantee the compatibility and the uniform boundedness, it is possible to statistically test/bound them by computing a high-probability upper bound on $\|F_\gamma^\sharp\|_2$ (*cf.* Proposition 19 and 20 in the appendix). Moreover, combining it with the non-asymptotic bound (Theorem 13 in the appendix), one can obtain a data-dependent concentration bound. If the resulting concentration rate is not acceptable, then one may change $\phi$ or fall back on tabular features.

### 3.2 Consistency of Tile-Coding Estimators under Unrealizability

As is seen from Theorem 6, not surprisingly, a linear estimator with fixed $\phi$ is not consistent in general under unrealizability. This motivates us to investigate alternative methods that adaptively selects feature mappings. Such an estimation method is formally defined as follows.

**Definition 11** (Nonparametric estimator). *We refer to $(\phi, \hat{m})$ as a nonparametric estimator if $\phi = \{\phi^{(m)}\}_{m=1}^{\infty}$ is a sequence of feature mappings such that $\phi^{(m)} : \mathcal{S} \times \mathcal{A} \to \mathbb{R}^{K_m}$, $K_m \geq 1$, and $\hat{m} : \mathbb{N} \to \mathbb{N}$ is a model-selecting function such that $\lim_{n \to \infty} \hat{m}(n) = \infty$. The output of the estimator is given as $\hat{J}(\pi; \phi, \hat{m}) := \hat{J}(\pi; \phi^{(\hat{m}(n))})$, where $\hat{J}(\pi; \phi)$ denotes the linear direct estimate given by Algorithm 1 with a feature mapping $\phi$.*

In principle, if $\hat{m}(n)$ diverges slowly compared to the convergence (6), the error of $(\phi, \hat{m})$ is still characterized by Theorem 6. Thus, if $\phi$ is such that the asymptotic bias given by (6) goes to zero as $m \to \infty$, there exists a consistent nonparametric estimator $(\phi, \hat{m})$ with sufficiently slowly diverging function $\hat{m}(n)$.

Below, we show a typical instance of such consistent nonparametric estimators, namely the refining tile-coding estimator. Henceforth, we assume for simplicity $\mathcal{S} \times \mathcal{A} = [0, 1]^d$, $d \geq 1$, i.e., both states and actions are continuous. [5]

**Definition 12** (Refining tile-coding sequence). *We call $\phi$ as the refining tile-coding sequence if, for all $m \geq 1$, $\phi^{(m)}$ is tabular with respect to the $m^d$-partition $\{\mathcal{P}_{\mathbf{k}}^{(m)}\}_{\mathbf{k} \in [m]^d}$ such that $\mathcal{P}_{\mathbf{k}}^{(m)} = \prod_{j=1}^{d} \mathcal{I}_{k_j}^{(m)}$ for all $\mathbf{k} = (k_1, ..., k_d) \in [m]^d$, where $\mathcal{I}_k^{(m)} := [\frac{k-1}{m}, \frac{k}{m})$ for all $1 \leq k < m$ and $\mathcal{I}_m^{(m)} := [1 - \frac{1}{m}, 1]$.*

Leveraging Theorem 13, the non-asymptotic version of Theorem 6, it is shown the nonparametric estimation using the tile-coding scheme is consistent under very mild assumptions.

**Proposition 10.** *For the refining tile-coding sequence $\phi$, we have $\hat{J}(\pi; \phi, \hat{m}) \overset{n \to \infty}{\longrightarrow} J(\pi)$ a.s. if*

1. $\Xi^n$ *is $G^*$-mixing for some $G^* < \infty$ (Assumption 1).*

2. $0 < c_\mu := \inf_{s \in \mathcal{S}, a \in \mathcal{A}} \mu(s, a) \leq C_\mu := \sup_{s \in \mathcal{S}, a \in \mathcal{A}} \mu(s, a) < \infty.$

3. $\hat{m}(n)^d / \sqrt{n} \to 0$ *as $n \to \infty$.*

Note that Proposition 10 assumes nothing on $p_r$, $p_T$, $p_0$ and $\pi$ other than the implicit well-definedness of their density functions. Because of this, the rate of convergence may be arbitrarily slow. We have a stronger guarantee if some regularities of the density ratio $\nu/\mu$ is given. A typical example of such regularity is the Lipschitz continuity.

**Proposition 11.** *Under the assumptions of Proposition 10, if $\nu/\mu$ is Lipschitz continuous on $\mathcal{S} \times \mathcal{A}$, then we have $|\hat{J}(\pi; \phi, \hat{m}) - J(\pi)| = \mathcal{O}(n^{-\frac{1}{2d+2}})$ with $\hat{m}(n) = \Theta(n^{\frac{1}{2d+1}})$.*

Note that the stronger convergence result is obtained still without any explicit conditions on the reward and the transition dynamics. This matches the implication of Theorem 6; the error can be controlled with the norm of $\mathcal{R}_\chi$, which measures the regularity of $\nu/\mu$, without forcing explicit regularities on the Bellman operator.

# 4   Related Work

The offline policy evaluation is closely related to the off-policy policy evaluation in the bandit and RL literatures (Precup, 2000; Dudík et al., 2011; Sutton and Barto, 2018), where behavior policies are often assumed to be known. Recently, a number of researchers are focusing on more 'offline' settings (Levine et al., 2020) featuring unknown data-collecting policies and relatively large distribution shifts.

The theory of OPE has been often studied under a number of different types of realizability conditions. A common type of realizability is the Bellman-operator realizability (Yin and Wang, 2020; Duan et al., 2020). Jin et al. (2020) also considered the same, but approximate realizability and presented an upper bound with respect to the violation of the realizability. Xie and Jiang (2020) studied offline RL under the Q-function realizability, a relaxation of the Bellman-operator realizability, also allowing small realizability-violation in their analysis. In comparison, we focus on exactly evaluating (dominating term of) the OPE error to understand more about the phenomenon under unrealizability, not just bounding it from above, which allows us to find the double robustness and show the consistency of the nonparametric estimator without realizability.

---

[5]It is possible to extend it to allow states and actions being discrete partially or entirely.

Doubly-robust properties similar to Theorem 6 were previously studied in the literature (Kallus and Uehara, 2020; Tang et al., 2019). In particular, Theorem 6 is quite similar to Theorem 3.1 in Tang et al. (2019) as both establish the double robustness of the bias of some OPE methods, although they are orthogonal to one another in the following senses. Theorem 6 shows the DR property of linear DM, while Kallus and Uehara (2020); Tang et al. (2019) establish those of generic meta algorithms based on given (nice) value-function and density-ratio estimators. In particular, Theorem 6 is nontrivial from the context of meta algorithm because linear DM does not perform any density-ratio estimation explicitly. Theorem 6 is also closely related to Example 2 and Example 7 in Uehara et al. (2020). These examples show that linear DM can be interpreted as both value function regression and density ratio regression, but their analysis requires *both* the Q-function *and* the density-ratio realizabilities to achieve the consistency (Table 1).

Theories of the kernel-based RL naturally takes into account the function approximation error. The difference between kernel-based methods and linear methods is subtle; Although kernel methods tends to be more expensive in computation, they can be approximated with a finite linear basis (Rahimi et al., 2007). Ormoneit and Sen (2002) shows a kernel-smoothing approach yields a nonparametric convergence rate of $\mathcal{O}(n^{-\frac{1}{2d+4}})$ under different regularity assumptions, including the continuity of $\bar{r}$. (Feng et al., 2020) studied a kernel-based OPE with confidence error bound, but the error bound is algorithmically determined and not studied analytically.

## 5 Concluding Remarks

We have derived an asymptotically exact characterization of the error of the linear direct estimators, one of the most simple and basic OPE methods, under a completely unrealizable setting. As a consequence, we have found that the error matches the inner product of two residual functions implying the double robustness of linear DM, each of which measures the unrealizability of the value function and the marginal density ratio, respectively. We have also found the compatibility of $\phi$ as a key condition that guarantees the above identity. Finally, we have investigated as its application the error of nonparametric estimators.

**Limitation.** One limitation of the present work is that we do not have any methods to control $\|F_\gamma^\sharp\|$ other than Proposition 9, which affects the speed of the concentration (6). In particular, this makes it difficult to construct consistent nonparametric estimators with general feature sequences $\phi$ other than the tabular ones since the norm $\|F_\gamma^\sharp\|$ may depend on $n$ in a nontrivial manner. However, as discussed in Section 3.1.1, controlling $\|F_\gamma^\sharp\|$ is at least equivalent of keeping the hard instances away from the easy instances in the sense of Wang et al. (2020) and Amortila et al. (2020). Thus, we speculate it could be a key question for better understanding of the hardness of OPE problems, rather than just a technical difficulty.

Another limitation is that all the results in this paper are derived under the implicit assumption that the marginal target density $\nu$ is well defined with respect to the base measure of $\mathcal{S} \times \mathcal{A}$. In particular, generalized functions like Dirac's delta density function cannot be handled straightforwardly in our framework.

**Feature Work.** The generalization of the double robustness of linear DM to nonlinear estimators would be an interesting future direction. Studies on the compatibility of the feature mappings $\phi$ with higher-order smoothness such as Gaussian radial basis functions, spline functions and neural tangent kernels (Jacot et al., 2018) are promising for faster convergence rate of the nonparametric estimator. Another promising direction is to extend the current analysis towards adaptive state-abstraction methods, e.g., Whiteson (2007).

## Acknowledgment

We deeply thank all the reviewers and the area chair for extensive discussions, which help us greatly improve the manuscript.

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
