# A  Proof of Theorem 6

Throughout the proof, we consider the general case where the concentrability (Assumption 3) does not necessarily hold, i.e., $\Sigma$ may be singular. Note that the projections $b^\sharp$, $F^\sharp$, $w^\sharp$ and $Q^\sharp$ may not be uniquely defined in this setting. Thus, we redefine them as the *least-norm* projections, where the norms are measured with $\|b^\sharp\|_2$, $\|F^\sharp\|_F$, $\|w^\sharp\|_{\mathcal{F}_\phi}$ and $\|Q^\sharp\|_{\mathcal{F}_\phi}$, respectively. Here, $\|A\|_F := \sqrt{\operatorname{tr} AA^\top}$ denotes the Frobenius norm of matrix and $\|f\|_{\mathcal{F}_\phi} := \inf_{\theta \in \mathbb{R}^K : \theta^\top \phi = f} \|\theta\|_2$ denotes the natural norm of $\mathcal{F}_\phi$. Accordingly, we repeatedly use the following notation.

**Definition 13** (Smallest nonzero eigenvalue and eivenvectors). *We denote by $c_* := 1/\lambda_1(\Sigma^+) > 0$ the smallest nonzero eigenvalue of $\Sigma$.*

*Moreover, let $\tilde{K} = \operatorname{rank} \Sigma$. We denote by $V \in \mathbb{R}^{K \times \tilde{K}}$ and $W^{K \times (K - \tilde{K})}$ the matrices of column eigenvectors corresponding to nonzero and zero eigenvalues, respectively. That is, $V$ and $W$ are column-maximal semi-orthogonal matrices satisfying*

$$\tilde{\Sigma} := V^\top \Sigma V \succ 0, \qquad\qquad \Sigma W = 0,$$

*and we have $c_* = \lambda_{\tilde{K}}(\tilde{\Sigma})$.*

We also frequently use the explicit formula of the projected Q-function $Q^\sharp$.

**Proposition 12.** *Suppose Assumption 4 holds. Let $\theta^\sharp := F_\gamma^{\sharp\top} b^\sharp$. Then, $\theta^\sharp \in \mathbb{R}^K$ is the least-norm parameter satisfying*

$$\theta^\sharp \in \operatorname*{argmin}_{\theta \in \mathbb{R}^K} \mathcal{L}_B(\theta^\top \phi),$$

*where $\mathcal{L}_B$ is given in Definition 7. Consequently, $Q^\sharp = \theta^{\sharp\top} \phi$.*

*Proof.* According to the stationary-point condition, any $\theta \in \operatorname{argmin}_{\theta \in \mathbb{R}^K} \mathcal{L}_B(\theta^\top \phi)$ satisfies

$$\begin{aligned}
0 &= \frac{\partial}{\partial \theta} |B(\theta^\top \phi) - \theta^\top \phi|_\phi^2 \\
&= 2\langle r\phi^\top + \theta^\top(\gamma P\phi - \phi)\phi^\top \rangle_\mu \left\langle \gamma \phi P \phi^\top - \phi\phi^\top \right\rangle_\mu \\
&= 2\left\{ (I - \gamma F^\sharp)^\top \theta - b^\sharp \right\}^\top \Sigma^2 (I - \gamma F^\sharp).
\end{aligned}$$

The least-norm solution to the above equation is given by $\theta = \theta^\sharp$. $\qquad\square$

To prove the main theorem, we exploit a more general form shown below. The notion of the characteristic numbers $(\epsilon, \bar{\alpha}, \bar{\beta})$ is newly introduced, whose definitions are given in order.

**Theorem 13.** *Let $0 < \delta < 1$ be a confidence parameter. Let $(\epsilon, \bar{\alpha}, \bar{\beta}) \in \mathbb{R}^3$ be the characteristic numbers associated with $(\mathcal{P}_{\mathrm{OPE}}, \phi)$, given by Definition 15. Under Assumption 1, 2 and 4,*

$$\left| \hat{J}(\pi) - J(\pi) - \frac{\epsilon}{1-\gamma} \right| \le \frac{1}{1-\gamma} \left\{ \left( \bar{\alpha} + \sqrt{2G^*}\bar{\beta} \right) \sqrt{\frac{2\ln\frac{12}{\delta}}{n}} \right\} +$$

$$\mathcal{O}\left( \frac{\bar{V}\rho G^* \ln^{3/4}\frac{2}{\delta}}{c_* n^{3/4}} + \frac{\bar{V}\rho^2 G^{*2} \ln\frac{2K}{\delta}}{c_*^2 n} \right). \qquad (7)$$

*with probability $1 - \delta$, where $n \ge \frac{512 G^{*2}\rho^2}{c_*^2} \ln(8K/\delta)$, $\rho := \left\|F_\gamma^\sharp\right\|_2$ and $\bar{V} := 1 + 2\left\|F_\gamma^{\sharp\top} b^\sharp\right\|_2$. Here, $\mathcal{O}(\cdot)$ is hiding a universal constant.*

*Proof.* (Sketch) Consider the policy value estimate of the projection, $J^\sharp(\pi) := b^{\sharp\top} F^\sharp x_0$, and the associated approximation-estimation error decomposition $\hat{J}(\pi) - J(\pi) = (J^\sharp(\pi) - J(\pi)) + (\hat{J}(\pi) - J^\sharp(\pi))$. The proof strategy is to evaluate these terms separately. In particular, the first term is exactly evaluated as $J^\sharp(\pi) - J(\pi) = \frac{\epsilon}{1-\gamma}$ and the second term is controlled with the martingale concentration inequalities. The full proof is deferred to Section B. $\qquad\square$

To define the characteristic numbers $(\epsilon, \bar{\alpha}, \bar{\beta})$, we need several auxiliary definitions.

**Definition 14** (Bellman bias and variance functions). *The Bellman bias function $\beta(s, a)$ and variance function $\alpha^2(s, a)$ is given by*

$$\beta(s, a) := \mathbb{E}[\mathcal{R}(s, a)], \qquad\qquad \alpha^2(s, a) := \mathbb{V}[\mathcal{R}(s, a)].$$

*where $\mathcal{R}(s, a) := Q^\sharp(s, a) - (R + \gamma Q^\sharp(S', A'))$ is a random function such that $(R, S', A') \sim p_r(r|s, a)p_T(s'|s, a)\pi(a'|s')$.*

Intuitively, the Bellman bias (variance) functions measures the bias (variance) in a single transition $(s, a, r, s')$ relative to the projection $(b^\sharp, F^\sharp)$, respectively. In particular, these functions are trivial if the environment $\mathcal{M}$ is realizable and/or deterministic; If $\mathcal{M} \in \mathcal{H}_\phi$, $\beta(\cdot, \cdot) \equiv 0$. Moreover, if $p_r(r|s, a)$ and $p_T(s'|s, a)$ are Dirac's delta functions, $\alpha^2(\cdot, \cdot) \equiv 0$.

Now we are prepared to state the definition of the characteristic numbers.

**Definition 15** (Characteristic numbers). *The characteristic numbers $(\epsilon, \bar{\alpha}, \bar{\beta})$ associated with $(\mathcal{P}_{\mathrm{OPE}}, \phi)$ is given by*

$$\epsilon := \langle \beta \rangle_\nu, \qquad\qquad \bar{\alpha} := \sqrt{\langle \sigma^2 \alpha^2 \rangle_\mu}, \qquad\qquad \bar{\beta} := \sqrt{\langle \sigma^2 \beta^2 \rangle_\mu},$$

*where $\sigma(s, a) := (1 - \gamma)x_0^\top (I - \gamma F^\sharp)^{-1\top}\Sigma^+ \phi(s, a)$.*

Now each of the characteristic numbers is evaluated as follows.

**Proposition 14.** *Under Assumption 2 and 4, we have*

$$\epsilon = -\langle \mathcal{R}_B \mathcal{R}_\chi \rangle_\mu, \qquad\qquad \bar{\alpha} \leq \frac{(1 - \gamma)\bar{V}\rho}{2\sqrt{c_*}}, \qquad\qquad \bar{\beta} \leq \frac{(1 - \gamma)\bar{V}\rho}{\sqrt{c_*}}.$$

*Proof.* Note that $b^\sharp = \Sigma^+ \langle \bar{r}\phi \rangle_\mu$ and $F^\sharp = \langle (P\phi)\phi^\top \rangle_\mu \Sigma^+$ by Corollary 22. Then, we have

$$
\begin{aligned}
Q^\sharp &:= b^{\sharp\top} F_\gamma^\sharp \phi \\
&= \left\{ b^{\sharp\top} + \gamma b^{\sharp\top} F_\gamma^\sharp F^\sharp \right\} \phi && (F_\gamma^\sharp = (I - \gamma F^\sharp)^{-1}) \\
&= \left\{ \langle \bar{r}\phi^\top \rangle_\mu \Sigma^+ + \gamma b^{\sharp\top} F_\gamma^\sharp \langle (P\phi)\phi^\top \rangle_\mu \Sigma^+ \right\} \phi \\
&= \left\{ \langle \bar{r}\phi^\top \rangle_\mu \Sigma^+ + \gamma \langle (PQ^\sharp)\phi^\top \rangle_\mu \Sigma^+ \right\} \phi \\
&= \langle BQ^\sharp \phi \rangle_\mu^\top \Sigma^+ \phi.
\end{aligned}
$$

According to Proposition 21, this implies $\mathcal{R}_B = BQ^\sharp - Q^\sharp$ lies in the orthogonal complement of $\mathcal{F}_\phi$ with respect to $\mu$, i.e., $\mathcal{R}_B \in \mathcal{F}_\phi^\perp := \{\tilde{f} : \mathcal{S} \times \mathcal{A} \to \mathbb{R} \mid \langle f\tilde{f} \rangle_\mu = 0, \forall f \in \mathcal{F}_\phi\}$. Now, since $\beta = -\mathcal{R}_B$ by definition,

$$
\begin{aligned}
\epsilon &= \langle \beta \rangle_\nu \\
&= -\langle \mathcal{R}_B \rangle_\nu \\
&= -\left\langle \mathcal{R}_B \frac{\nu}{\mu} \right\rangle_\mu \\
&= -\left\langle \mathcal{R}_B \left( \frac{\nu}{\mu} - f^* \right) \right\rangle_\mu && (\mathcal{R}_B \in \mathcal{F}_\phi^\perp)
\end{aligned}
$$

for any $f^* \in \mathcal{F}_\phi$. Taking $f^* = w^\sharp$ as in Definition 9 proves the equality of $\epsilon$.

Now observe

$$\|\alpha\|_\infty \leq \frac{\bar{V}}{2}, \qquad\qquad \|\beta\|_\infty \leq \bar{V},$$

since $|\mathcal{R}(s,a)| = |Q^\sharp(s,a) - R - \gamma Q^\sharp(S',A')| \leq 1 + 2\|Q^\sharp\|_\infty \leq \bar{V}$. Thus the remaining inequalities are proved by

$$
\begin{aligned}
\left\langle \sigma^2 \right\rangle_\mu &= (1-\gamma)^2 \left\langle \left( x_0^\top (I - \gamma F^\sharp)^{-1\top} \Sigma^+ \phi \right)^2 \right\rangle_\mu \\
&= (1-\gamma)^2 x_0^\top (I - \gamma F^\sharp)^{-1\top} \Sigma^+ \left\langle \phi\phi^\top \right\rangle_\mu \Sigma^+ (I - \gamma F^\sharp)^{-1} x_0 \\
&= (1-\gamma)^2 x_0^\top (I - \gamma F^\sharp)^{-1\top} \Sigma^+ (I - \gamma F^\sharp)^{-1} x_0 \\
&\leq (1-\gamma)^2 \rho^2 \left\|\Sigma^+\right\|_2 && (\|x_0\| \leq 1) \\
&\leq (1-\gamma)^2 \rho^2 \frac{1}{c_*} && \text{(Definition 13)}
\end{aligned}
$$

$\square$

Now, combining Theorem 13 and Proposition 14, we get

$$
\left| \hat{J}(\pi) - J(\pi) - \frac{\epsilon}{1-\gamma} \right| \leq \frac{\bar{V}\rho\left(1 + 2\sqrt{2G^*}\right)}{2\sqrt{c_*}} \sqrt{\frac{2\ln\frac{12}{\delta}}{n}} + \mathcal{O}\left( \frac{\bar{V}\rho G^* \ln^{3/4} \frac{2}{\delta}}{c_* n^{3/4}} + \frac{\bar{V}\rho^2 G^{*2} \ln\frac{2K}{\delta}}{c_*^2 n} \right).
\tag{8}
$$

Noting that the RHS of (8) is independent of $p_0$, we obtain the first desired result, i.e., the $p_0$-uniform convergence under the existence of $F_\gamma^\sharp$, by the Borel–Cantelli lemma.

Finally, to show the $\pi$-uniform convergence, we need to uniformly bound the $\pi$-dependent quantities on the RHS, i.e., $\rho$ and $\bar{V}$. As for $\sup_\pi \rho$, it is directly bounded by the assumption of Theorem 6. As for $\sup_\pi \bar{V}$, observe

$$
\begin{aligned}
\bar{V} &= 1 + 2\left\| F_\gamma^{\sharp\top} b^\sharp \right\|_2 \\
&\leq 1 + 2\rho \left\| b^\sharp \right\|_2 \\
&= 1 + 2\rho \left\| \Sigma^+ \left\langle \bar{r}\phi \right\rangle_\mu \right\|_2 && \text{(Proposition 22)} \\
&\leq 1 + \frac{2\rho}{c_*} \left\| \left\langle \bar{r}\phi \right\rangle_\mu \right\|_2 \\
&\leq 1 + \frac{2\rho}{c_*} \left\langle \|\bar{r}\phi\|_2 \right\rangle_\mu \\
&\leq 1 + \frac{2\rho}{c_*}, && \text{(Boundedness of reward and Assumption 2)}
\end{aligned}
$$

where the last expression is uniformly bounded as long as $\sup_\pi \rho$ is uniformly bounded. This concludes the proof.

## B  Proof of Theorem 13

Consider the decomposition $\hat{J}(\pi) - J(\pi) = (J^\sharp(\pi) - J(\pi)) + (\hat{J}(\pi) - J^\sharp(\pi))$, where $J^\sharp(\pi)$ is defined as follows.

**Definition 16.** *Let $J^\sharp(\pi)$ denote the projected policy value given by $J^\sharp(\pi) := \left\langle Q^\sharp \right\rangle_{p_0^\pi} = b^{\sharp\top} F_\gamma^\sharp x_0$.*

Then, the approximation error $J^\sharp(\pi) - J(\pi)$ is evaluated as follows.

**Lemma 15.**

$$
J^\sharp(\pi) - J(\pi) = \frac{\epsilon}{1-\gamma}.
$$

*Proof.* Note that $J^{\sharp}(\pi) = \langle Q^{\sharp} \rangle_{p_0^{\pi}}$ and $J(\pi) = \langle (I - \gamma P)^{-1} \bar{r} \rangle_{p_0^{\pi}}$. Therefore,

$$
\begin{aligned}
J^{\sharp}(\pi) - J(\pi) &= \langle (I - \gamma P)^{-1}((I - \gamma P)Q^{\sharp} - \bar{r}) \rangle_{p_0^{\pi}} \\
&= \langle (I - \gamma P)^{-1} \beta \rangle_{p_0^{\pi}} \\
&= \frac{1}{1 - \gamma} \langle \beta \rangle_{\nu} \qquad\qquad\qquad (\because \nu = (1 - \gamma)(I - \gamma P^{\dagger})^{-1} p_0^{\pi}) \\
&= \frac{\epsilon}{1 - \gamma}.
\end{aligned}
$$

$\square$

On the other hand, the estimation error $\hat{J}(\pi) - J^{\sharp}(\pi)$ is bounded as follows, which concludes the proof.

**Lemma 16.** *Let* $0 < \delta < 1$. *Under Assumption 1 and 2, if* $\rho := \left\| F_{\gamma}^{\sharp} \right\|_2 < \infty$ *and* $n \geq \frac{512 G^{*2} \rho^2}{c_*^2} \ln(8K/\delta)$,

$$
\left| \hat{J}(\pi) - J^{\sharp}(\pi) \right| \leq \frac{\bar{\alpha} + \sqrt{2 G^*} \bar{\beta}}{1 - \gamma} \sqrt{\frac{2 \ln(12/\delta)}{n}} + \mathcal{O}\left( \frac{\bar{V} \rho G^* \ln^{3/4} \frac{2}{\delta}}{c_* n^{3/4}} + \frac{\bar{V} \rho^2 G^{*2} \ln \frac{2K}{\delta}}{c_*^2 n} \right). \qquad (9)
$$

*with probability* $1 - \delta$.

*Proof.* Consider the event $E$ wherein the inequalities of Proposition 17, 18 and 20 are simultaneously true. Note $\mathbb{P}(E) \geq 1 - 2\delta$ and suppose $E$ occurs.

By Proposition 20, $\hat{F}_{\gamma} := (I - \gamma \hat{F})^{-1}$ is well-defined under $E$. Let $\theta^{\sharp} = F_{\gamma}^{\sharp \top} b^{\sharp}$ and observe that

$$
\begin{aligned}
\hat{J}(\pi) - J^{\sharp}(\pi) &= \hat{b}^{\top} \hat{F}^{\sharp} x_0 - b^{\sharp \top} F_{\gamma}^{\sharp} x_0 \\
&= (\hat{b} - b^{\sharp})^{\top} \hat{F}^{\sharp} x_0 + b^{\sharp \top}(\hat{F}_{\gamma} - F_{\gamma}^{\sharp}) x_0 \\
&= (\hat{b} - b^{\sharp})^{\top} \hat{F}^{\sharp} x_0 + \gamma b^{\sharp \top} F_{\gamma}^{\sharp}(\hat{F} - F^{\sharp}) \hat{F}_{\gamma} x_0 \\
&= (\hat{b} + \gamma \hat{F}^{\top} \theta^{\sharp} - \theta^{\sharp})^{\top} \hat{F}_{\gamma} x_0 \qquad\qquad (b^{\sharp} + \gamma F^{\sharp \top} \theta^{\sharp} = \theta^{\sharp}). \qquad (10)
\end{aligned}
$$

Moreover, since

$$
\begin{aligned}
\hat{b} - b^{\sharp} &= \frac{1}{n} \hat{\Sigma}^{+} \Phi^{\top} \bar{r} - b^{\sharp} \\
&= \frac{1}{n} \hat{\Sigma}^{+} \Phi^{\top}(\bar{r} - \Phi b^{\sharp}),
\end{aligned}
$$

and

$$
\begin{aligned}
\hat{F} - F^{\sharp} &= \frac{1}{n} \Psi_{\pi}^{\top} \Phi \hat{\Sigma}^{+} - F^{\sharp} \\
&= \frac{1}{n}(\Psi_{\pi}^{\top} - F^{\sharp} \Phi^{\top}) \Phi \hat{\Sigma}^{+}, \qquad\qquad (11)
\end{aligned}
$$

we have

$$
\hat{J}(\pi) - J^{\sharp}(\pi) = \hat{z}^{\top} \hat{\Sigma}^{+} \hat{F}_{\gamma} x_0,
$$

where $\hat{z} := \frac{1}{n} \Phi^{\top} \left\{ \bar{r} + \gamma \Psi_{\pi} \theta^{\sharp} - \Phi \theta^{\sharp} \right\}$.

Now according to Proposition 17, 18 and 20, under $E$,

$$
\| \hat{z} \|_2 \leq 2 \bar{V} G^* \sqrt{\frac{2 \ln \frac{4K}{\delta}}{n}},
$$

$$
\left\| \hat{\Sigma}^{+} - \Sigma^{+} \right\|_2 \leq \frac{4 G^*}{c_*^2} \sqrt{\frac{2 \ln \frac{4K}{\delta}}{n}},
$$

$$
\left\| \hat{F}_{\gamma} - F_{\gamma}^{\sharp} \right\|_2 \leq \frac{16 G^* \rho^2}{c_*} \sqrt{\frac{2 \ln \frac{8K}{\delta}}{n}},
$$

which implies $\hat{J}(\pi) - J^\sharp(\pi)$ is well approximated by $Z := \hat{z}^\top \Sigma^+ F_\gamma^\sharp x_0$ with probability $1 - 2\delta$,

$$
\begin{aligned}
\left| \hat{J}(\pi) - J^\sharp(\pi) - Z \right| &= \left| \hat{z}^\top (\hat{\Sigma}^+ \hat{F}_\gamma - \Sigma^+ F_\gamma^\sharp) x_0 \right| \\
&\leq \|\hat{z}\|_2 \left( \|\Sigma^+\|_2 + \left\| \hat{\Sigma}^+ - \Sigma^+ \right\|_2 \right) \left( \|F_\gamma^\sharp\|_2 + \left\| \hat{F}_\gamma - F_\gamma^\sharp \right\|_2 \right) \\
&\leq \|\hat{z}\|_2 \left( c_*^{-1} + \left\| \hat{\Sigma}^+ - \Sigma^+ \right\|_2 \right) \left( \rho + \left\| \hat{F}_\gamma - F_\gamma^\sharp \right\|_2 \right) \\
&= \mathcal{O}\left( \frac{\bar{V} G^{*2} \|F_\gamma^\sharp\|_2^2 \ln \frac{2K}{\delta}}{c_*^2 n} \right),
\end{aligned}
\tag{12}
$$

where $\mathcal{O}(\cdot)$ is hiding a universal constant.

To bound $Z$, note that $Z = \frac{1}{(1-\gamma)n} \sum_{i=1}^n \varepsilon(\Xi_i)\, \sigma(S_i, A_i)$, where $\varepsilon(\xi_i) := r_i + \gamma \theta^{\sharp\top} \psi_\pi(s_i') - \theta^{\sharp\top} \phi(s_i, a_i)$. Thus, letting $f(\xi_i) = \frac{1}{(1-\gamma)n} \varepsilon(\xi_i)\sigma(s_i, a_i)$, we learn from Proposition 26

$$
\begin{aligned}
|Z - \mathbb{E}[Z]| &\leq \sqrt{2\mathbb{V}[Z] \ln(4/\delta)} + \mathcal{O}\left( G^* \|f\|_\infty\, n^{1/4} \ln^{3/4}(2/\delta) \right) \\
&= \sqrt{2\mathbb{V}[Z] \ln(4/\delta)} + \mathcal{O}\left( \frac{\bar{V} G^* \|F_\gamma^\sharp\|_2}{c_* n^{3/4}} \ln^{3/4}(2/\delta) \right).
\end{aligned}
\tag{13}
$$

Now the proof is completed by evaluating $\mathbb{E}[Z]$ and $\mathbb{V}[Z]$. Observe $\mathbb{E}[\hat{z}] = \langle -\beta\phi \rangle_\mu = 0$ from Proposition 23 and thus

$$
\mathbb{E}[Z] = \mathbb{E}[\hat{z}]^\top \Sigma^+ F_\gamma^\sharp x_0 = 0.
$$

To bound $\mathbb{V}[Z]$, consider the decomposition

$$
\begin{aligned}
Z &= Z_\alpha + Z_\beta, \\
Z_\alpha &:= \frac{1}{(1-\gamma)n} \sum_{i=1}^n (\varepsilon_i - \beta(S_i, A_i))\, \sigma(S_i, A_i), \\
Z_\beta &:= \frac{1}{(1-\gamma)n} \sum_{i=1}^n \beta(S_i, A_i)\, \sigma(S_i, A_i),
\end{aligned}
$$

which yields an upper bound $\sqrt{\mathbb{V}[Z]} \leq \sqrt{\mathbb{V}[Z_\alpha]} + \sqrt{\mathbb{V}[Z_\beta]}$.

The first term $\mathbb{V}[Z_\alpha]$ is evaluated straightforwardly since $\mathbb{E}[(\varepsilon_i - \beta(S_i, A_i))^2 | \Xi^i] = \alpha^2(S_i, A_i)$,

$$
\begin{aligned}
\mathbb{V}[Z_\alpha] &= \mathbb{E}\left( \frac{1}{(1-\gamma)n} \sum_{i=1}^n (\varepsilon_i - \beta(S_i, A_i))\, \sigma(S_i, A_i) \right)^2 \\
&= \frac{1}{(1-\gamma)^2 n^2} \sum_{i=1}^n \mathbb{E}\left[ (\varepsilon_i - \beta(S_i, A_i))^2\, \sigma^2(S_i, A_i) \right] \\
&= \frac{1}{(1-\gamma)^2 n^2} \sum_{i=1}^n \mathbb{E}\left[ (\sigma^2 \alpha^2)(S_i, A_i) \right] \\
&= \frac{\langle \sigma^2 \alpha^2 \rangle_\mu}{(1-\gamma)^2 n} \\
&= \frac{\bar{\alpha}^2}{(1-\gamma)^2 n}.
\end{aligned}
$$

On the other hand, the second term $\mathbb{V}[Z_\beta]$ is bounded by Proposition 24,

$$
\begin{aligned}
\mathbb{V}[Z_\beta] = \mathbb{V}\left[ \frac{1}{(1-\gamma)n} \sum_{i=1}^{n} \beta(S_i, A_i)\, \sigma(S_i, A_i) \right] \\
\leq 2G^* \sum_{i=1}^{n} \mathbb{V}\left[ \frac{1}{(1-\gamma)n} \beta(S_i, A_i)\, \sigma(S_i, A_i) \right] \\
= \frac{2G^*}{(1-\gamma)^2 n^2} \sum_{i=1}^{n} \mathbb{V}\left[ \beta(S_i, A_i)\, \sigma(S_i, A_i) \right] \\
\leq \frac{2G^*}{(1-\gamma)^2 n^2} \sum_{i=1}^{n} \mathbb{E}\left[ (\sigma^2 \beta^2)(S_i, A_i) \right] \\
= \frac{2G^* \left\langle \sigma^2 \beta^2 \right\rangle_\mu}{(1-\gamma)^2 n} \\
= \frac{2G^* \bar{\beta}^2}{(1-\gamma)^2 n}.
\end{aligned}
$$

Back to (13), we get

$$
\begin{aligned}
Z &\leq \sqrt{2\mathbb{V}[Z]\ln(4/\delta)} + \mathcal{O}\left( \frac{\bar{V} G^* \left\| F_\gamma^\sharp \right\|_2}{c_* n^{3/4}} \ln^{3/4}(2/\delta) \right) \\
&\leq \frac{\bar{\alpha} + \sqrt{2G^*}\bar{\beta}}{(1-\gamma)} \sqrt{\frac{2\ln(4/\delta)}{n}} + \mathcal{O}\left( \frac{\bar{V} G^* \left\| F_\gamma^\sharp \right\|_2}{c_* n^{3/4}} \ln^{3/4}(2/\delta) \right)
\end{aligned}
$$

with probability $1 - \delta$. Combining it with (12) and take $\delta \leftarrow \delta/3$ concludes the proof. $\qquad\square$

## C Individual Concentration Bounds

### C.1 Concentration of $\|\hat{z}\|_2$

**Proposition 17.** *Let* $\hat{z} \coloneqq \frac{1}{n}\Phi^\top \left\{ \bar{r} + \gamma \Psi_\pi \theta^\sharp - \Phi \theta^\sharp \right\}$ *and* $\bar{V} \coloneqq 1 + 2\left\| F_\gamma^{\sharp\top} b^\sharp \right\|_2$. *Then, under Assumption 1 and 2,*

$$
\|\hat{z}\|_2 \leq 2\bar{V} G^* \sqrt{\frac{2\ln\frac{4K}{\delta}}{n}}
$$

*with probability* $1 - \delta$.

*Proof.* Observe

$$
\hat{z} = \frac{1}{n} \sum_{i=1}^{n} \tilde{\mathcal{R}}(\Xi_i)\phi(S_i, A_i),
$$

where $\tilde{\mathcal{R}}(\xi_i) \coloneqq r_i + \gamma\theta^{\sharp\top}\psi_\pi(s_i') - \theta^{\sharp\top}\phi(s_i, a_i)$. Note $\mathbb{E}[\tilde{\mathcal{R}}(\Xi_i)|S_i = s, A_i = a] = \mathbb{E}[\mathcal{R}(s,a)] = \beta(s,a)$ for $1 \leq i \leq n$ (Definition 14), which implies $\hat{z}$ is centered, $\mathbb{E}[\hat{z}] = \left\langle \beta\phi \right\rangle_\mu = 0$, according to Proposition 23. Moreover, each summand is almost surely bounded by $\left\| \tilde{\mathcal{R}}(\Xi_i)\phi(S_i, A_i) \right\|_2 \leq \left| \tilde{\mathcal{R}}(\Xi_i) \right| \leq \bar{V}$, $1 \leq i \leq n$.

Now consider the Doob martingale of $\hat{z}$, $Y_j \coloneqq \mathbb{E}[\hat{z}|\Xi^j]$, $0 \leq j \leq n$. Then, Proposition 25 with $f(\xi_i) = \tilde{\mathcal{R}}(\xi_i)\phi(s_i, a_i)/n$ yields the desired result, where $\|f\|_\infty = \bar{V}$. $\qquad\square$

## C.2 Concentration of $\hat{\Sigma}$.

**Proposition 18.** *Under Assumption 1 and 2, we have*

$$\left\|\hat{\Sigma} - \Sigma\right\|_2 \leq 2G^* \sqrt{\frac{2\ln\frac{4K}{\delta}}{n}} \tag{14}$$

*with probability $1 - \delta$. Moreover, we also have*

$$\left\|\hat{\Sigma}^+ - \Sigma^+\right\|_2 \leq \frac{4G^*}{c_*^2} \sqrt{\frac{2\ln\frac{4K}{\delta}}{n}}$$

*if $n \geq \frac{16G^{*2}}{c_*^2} \ln(4K/\delta)$.*

*Proof.* Consider the Doob martingale given by $Y_j := \mathbb{E}[\hat{\Sigma}|\Xi^j]$, $0 \leq j \leq n$. Then Proposition 25 yields the first inequality, where $f(\xi_i) = \phi(s_i, a_i)\phi(s_i, a_i)^\top$ and $\|f\|_\infty \leq 1$ (Assumption 2).

To prove the second inequality, observe that the first inequality with $n \geq \frac{16G^{*2}}{c_*^2} \ln(4K/\delta)$ implies $\|\hat{\Sigma} - \Sigma\|_2 \leq c_*/2$. Therefore, we have $V^\top \hat{\Sigma} V \succ 0$. The concentration of the pseudo-inverse is proved by

$$
\begin{aligned}
\|\hat{\Sigma}^+ - \Sigma^+\|_2 &= \|\Sigma^+(\Sigma - \hat{\Sigma})\hat{\Sigma}^+\|_2 && (\Sigma^+\hat{\Sigma}\hat{\Sigma}^+ = \Sigma^+ \text{ since } V^\top\hat{\Sigma}V \succ 0) \\
&\leq \|\Sigma^+\|_2\|\hat{\Sigma}^+\|_2\|\hat{\Sigma} - \Sigma\|_2 \\
&\leq \frac{\|\hat{\Sigma} - \Sigma\|_2}{c_*(c_* - \|\hat{\Sigma} - \Sigma\|_2)} && (V^\top\hat{\Sigma}V \succ 0) \\
&\leq \frac{2\|\hat{\Sigma} - \Sigma\|_2}{c_*^2}. && (\|\hat{\Sigma} - \Sigma\|_2 \leq c_*/2)
\end{aligned}
$$

This concludes the proof. $\qquad\square$

## C.3 Concentration of $\hat{F}_\gamma$.

**Proposition 19.** *Under Assumption 1 and 2, if $n \geq \frac{16G^{*2}}{c_*^2} \ln(8K/\delta)$, we have*

$$\left\|\hat{F} - F^\sharp\right\|_2 \leq \frac{8G^*}{c_*} \sqrt{\frac{2\ln\frac{8K}{\delta}}{n}}$$

*in addition to the inequalities of Proposition 18, with probability $1 - \delta$.*

*Proof.* Proposition 18 shows $\|\hat{\Sigma}^+\|_2 = 1/\lambda_K(\hat{\Sigma}) \leq (c_* - \|\hat{\Sigma} - \Sigma_n\|_2)^{-1} \leq 2/c_*$ with probability $1 - \delta/2$. Thus, combining it with the identity

$$\hat{F} - F^\sharp = \frac{1}{n}\Psi_\pi^\top \Phi \hat{\Sigma}^+ - F^\sharp = \frac{1}{n}(\Psi_\pi^\top - F^\sharp \Phi^\top)\Phi\hat{\Sigma}^+,$$

we have

$$\left\|\hat{F} - F^\sharp\right\|_2 \leq \frac{2}{nc_*} \|Y_n\|_2. \tag{15}$$

where $Y_j := \mathbb{E}[(\Psi_\pi^\top - F\Phi^\top)\Phi|\Xi^j]$, $0 \leq j \leq n$. Now, let $f(\xi_i) = (\psi_\pi(s_i') - F^\sharp\phi(s_i, a_i))\phi(s_i, a_i)^\top$ and note $\|f\|_\infty \leq 2$ by Assumption 2. Note also $Y_0 = n\left\langle(P\phi - F\phi)\phi^\top\right\rangle_\mu = 0$ owing to Proposition 23. Thus Proposition 25 yields

$$\|Y_n\|_2 \leq 4G^* \sqrt{2n\ln\frac{8K}{\delta}} \tag{16}$$

with probability $1 - \delta/2$. This completes the proof. $\qquad\square$

**Proposition 20.** *Under Assumption 1 and 2, if $\rho := \left\| F_\gamma^\sharp \right\|_2 < \infty$ and $n \geq \frac{512 G^{*2} \rho^2}{c_*^2} \ln(8K/\delta)$, we have*

$$\left\| \hat{F}_\gamma - F_\gamma^\sharp \right\|_2 \leq \frac{16 G^* \rho^2}{c_*} \sqrt{\frac{2 \ln \frac{8K}{\delta}}{n}}$$

*in addition to the inequalities of Proposition 18 and 19, with probability $1 - \delta$.*

*Proof.* Consider the event $E$ wherein Proposition 18 and 19 are simultaneously true. Note $\mathbb{P}(E) \geq 1 - \delta$ and suppose $E$ occurs.

Observe

$$I - \gamma \hat{F} = (I - \gamma F^\sharp)(I + \gamma F_\gamma^\sharp (\hat{F} - F^\sharp)),$$

where $F_\gamma^\sharp = (I - \gamma F^\sharp)^{-1}$. Since we have $\|\hat{F} - F^\sharp\|_2 \leq \frac{1}{2\rho}$ (under $E$), it follows that

$$\left\| F_\gamma^\sharp (\hat{F} - F^\sharp) \right\|_2 \leq \rho \left\| \hat{F} - F^\sharp \right\|_2 \leq 1/2,$$

which ensures the existence of $(I + \gamma F_\gamma^\sharp (\hat{F} - F^\sharp))^{-1}$, and therefore the existence of $\hat{F}_\gamma := (I - \gamma \hat{F})^{-1}$. Moreover,

$$\begin{aligned}
\left\| \hat{F}_\gamma - F_\gamma^\sharp \right\|_2 &= \left\| \gamma \hat{F}_\gamma (\hat{F} - F^\sharp) F_\gamma^\sharp \right\|_2 \\
&\leq \gamma \rho \left\| \hat{F} - F^\sharp \right\|_2 \left\| \hat{F}_\gamma \right\|_2 \qquad (17) \\
&\leq \frac{1}{2} \left\| \hat{F}_\gamma \right\|_2.
\end{aligned}$$

Thus, the triangle inequality $\|\hat{F}_\gamma\|_2 \leq \|F_\gamma^\sharp\|_2 + \|\hat{F}_\gamma - F_\gamma^\sharp\|_2$ shows that $\|\hat{F}_\gamma\|_2 \leq 2\|F_\gamma^\sharp\|_2 = 2\rho$ Putting it back to (17), we get

$$\left\| \hat{F}_\gamma - F_\gamma^\sharp \right\|_2 \leq 2\gamma \rho^2 \left\| \hat{F} - F^\sharp \right\|_2.$$

Finally, Proposition 19 under $E$ yields the first desired result.

$\square$

## D  Additional Definitions

**Definition 17** ('$\phi$'-mixing coefficient, Bradley (2005)). *The '$\phi$'-mixing coefficients of $\Xi^n$, $g(h)$, is defined as*

$$g(h) := \sup_{1 \leq i \leq n-h} \sup_{D_1 \subset \mathcal{D}^i} \sup_{D_2 \subset \mathcal{D}^{n-i-h+1}} |\mathbb{P}(\Xi_{\geq i+h} \in D_2 \mid \Xi_{\leq i} \in D_1) - \mathbb{P}(\Xi_{\geq i+h} \in D_2)|$$

*for all $1 \leq h \leq n - 1$ and $g(0) = 1$, where $D_1$ and $D_2$ range over the measurable sets. Here, $\Xi_{\leq i} := (\Xi_1, ..., \Xi_i)$ and $\Xi_{\geq j} := (\Xi_j, ..., \Xi_n)$, respectively for all $1 \leq i, j \leq n$.*

## E  Additional Propositions

### E.1  Explicit Formula for Least Squares Projections

**Proposition 21.** *Let $f$ be an arbitrary $\mathbb{R}^{K'}$-valued function defined on $\mathcal{S} \times \mathcal{A}$, $K' \in \mathbb{N}$. Let $A^* \in \mathbb{R}^{K' \times K}$ be the least-Frobenius-norm solution to*

$$\min_{A \in \mathbb{R}^{K' \times K}} \left\langle \|f - A\phi\|_2^2 \right\rangle_\mu.$$

*Then, we have*

$$A^* = \left\langle f \phi^\top \right\rangle_\mu \Sigma^+$$

*and thus, for any $\tilde{f}(s, a) = v^\top \phi(s, a)$, $s \in \mathcal{S}$, $a \in \mathcal{A}$,*

$$\left\langle (f - A^* \phi) \tilde{f} \right\rangle_\mu = 0.$$

*Proof.* The solution $A^*$ satisfies the stationary point condition,

$$
\begin{aligned}
0 &= \frac{\partial}{\partial A} \left\langle \|f - A\phi\|_2^2 \right\rangle_\mu \\
&= \left\langle \frac{\partial}{\partial A} \|f - A\phi\|_2^2 \right\rangle_\mu \\
&= 2 \left\langle (f - A\phi)\phi^\top \right\rangle_\mu \\
&= 2 \left( \left\langle f\phi^\top \right\rangle_\mu - A \left\langle \phi\phi^\top \right\rangle_\mu \right) \\
&= 2 \left( \left\langle f\phi^\top \right\rangle_\mu - A\Sigma \right).
\end{aligned}
$$

Thus, any stationary points can be written as $A^* = \left\langle f\phi^\top \right\rangle_\mu \Sigma^+ + XW^\top$ for some $X \in \mathbb{R}^{K \times (K-\tilde{K})}$. The least norm is attained if and only if $X = 0$. $\qquad\square$

**Corollary 22.** *Let $(b^\sharp, F^\sharp)$ be given as in Definition 4. Then, we have the following.*

$$
b^\sharp = \Sigma^+ \left\langle \bar{r}\phi \right\rangle_\mu, \qquad\qquad F^\sharp = \left\langle (P\phi)\phi^\top \right\rangle_\mu \Sigma^+,
$$

*Proof.* It immediately follows from Proposition 21. $\qquad\square$

### E.2  Zero-Bias Projection

**Proposition 23.** *We have*

$$
\left\langle (\bar{r} - b^{\sharp\top}\phi)\phi^\top \right\rangle_\mu = 0, \qquad \left\langle (P\phi - F^\sharp\phi)\phi^\top \right\rangle_\mu = 0, \qquad \left\langle \beta\phi \right\rangle_\mu = 0.
$$

*Proof.* By Definition 4, the stationary-point condition on $(b^\sharp, F^\sharp)$ is given as

$$
0 = \mathbb{E}[\Phi^\top \hat{r}] - n\Sigma b^\sharp = n \left\langle (\bar{r} - b^{\sharp\top}\phi)\phi^\top \right\rangle_\mu, \tag{18}
$$

$$
0 = \mathbb{E}[\Psi_\pi^\top \Phi] - nF^\sharp\Sigma = n \left\langle (P\phi - F^\sharp\phi)\phi^\top \right\rangle_\mu, \tag{19}
$$

which proves the first two equalities. To see the last equality, observe

$$
\beta(s,a) = \bar{r}(s,a) - b^{\sharp\top}\phi(s,a) + \gamma\theta^{\sharp\top} \left\{ (P\phi)(s,a) - F^\sharp\phi(s,a) \right\}
$$

by Definition 14, where $\theta^\sharp := F_\gamma^\sharp b^\sharp$. Thus, adding (18) and (19) multiplied with $\gamma\theta^{\sharp\top}$ from left, we get the desired result. $\qquad\square$

### E.3  Concentration Bounds under $G^*$-mixing

**Proposition 24** (Variance bound)**.** *Let $f$ be a $\mathbb{R}$-valued function on $\mathcal{D}$ and $Y := \sum_{i=1}^n f(\Xi_i)$. Then, under Assumption 1,*

$$
\mathbb{V}[Y|\Xi^j] \leq 2G^* \sum_{i=j+1}^n \mathbb{V}[f(\Xi_i)|\Xi^j]
$$

*for all $0 \leq j \leq n$.*

*Proof.* Let $\mathrm{Cov}_j[\cdot, \cdot]$ denote $\mathrm{Cov}[\cdot, \cdot \,|\, \Xi^j]$ and let $f_i := f(\Xi_i)$. Then

$$
\begin{aligned}
\mathbb{V}_j[Y] &= \mathbb{V}_j \left[ \sum_{i=1}^n f_i \right] \\
&= \mathbb{V}_j \left[ \sum_{i=j+1}^n f_i \right] \\
&= \sum_{i=j+1}^n \sum_{\ell=j+1}^n \mathrm{Cov}_j[f_i, f_\ell].
\end{aligned}
$$

Note that $\text{Cov}_j[f_i, f_\ell] \le 2\sqrt{g(|i - \ell|)\mathbb{V}_j[f_i]\mathbb{V}_j[f_\ell]}$ by the domination of the '$\phi$'-mixing over the '$\rho$'-mixing, Eq. (1.13) in Bradley (2005), which implies

$$
\begin{aligned}
\mathbb{V}_j[Y] &= \sum_{i=j+1}^{n} \sum_{\ell=j+1}^{n} \text{Cov}_j[f_i, f_\ell] \\
&\le \sum_{i=j+1}^{n} \sum_{\ell=j+1}^{n} 2\sqrt{g(|i - \ell|)\mathbb{V}_j[f_i]\mathbb{V}_j[f_\ell]} \\
&\le \sum_{i=j+1}^{n} \sum_{\ell=j+1}^{n} \sqrt{g(|i - \ell|)} \left\{\mathbb{V}_j[f_i] + \mathbb{V}_j[f_\ell]\right\} \qquad \text{(AM-GM inequality)} \\
&= 2 \sum_{i=j+1}^{n} \mathbb{V}_j[f_i] \sum_{\ell=j+1}^{n} \sqrt{g(|i - \ell|)} \\
&\le 2G^* \sum_{i=j+1}^{n} \mathbb{V}_j[f_i] \qquad \text{(Assumption 1)}
\end{aligned}
$$

$\square$

**Proposition 25** (Hoeffding type, matrix form). *Let $f$ be a $\mathbb{R}^{K \times K'}$-valued function on $\mathcal{D}$ and let $Y := \sum_{i=1}^{n} f(\Xi_i)$. Then, under Assumption 1,*

$$
\|Y - \mathbb{E}[Y]\|_2 \le 2G^* \|f\|_\infty \sqrt{2n \ln \frac{4(K \vee K')}{\delta}}
$$

*with probability $1 - \delta$, where $\|f\|_\infty := \text{ess sup}_{\xi \in \mathcal{D}} \|f(\xi)\|_2$. Here, $a \vee b$ denotes $\max\{a, b\}$.*

*Proof.* Let $Y_j := \mathbb{E}[Y | \Xi^j]$. Note $Y_j$ is a matrix martingale with difference bounded by

$$
\begin{aligned}
&\|Y_j - Y_{j-1}\|_2 \\
&\le \sum_{i=j}^{n} \left\|\mathbb{E}[f(\Xi_i)|\Xi^j] - \mathbb{E}[f(\Xi_i)|\Xi^{j-1}]\right\|_2 \\
&\le \sum_{i=j}^{n} 2\|f\|_\infty \sup_{D \subset \mathcal{D}} \left|\mathbb{P}(\Xi_i \in D|\Xi^j) - \mathbb{P}(\Xi_i \in D|\Xi^{j-1})\right| \\
&\le 2\|f\|_\infty \sum_{i=j}^{n} \left\{\sup_{D \subset \mathcal{D}} \left|\mathbb{P}(\Xi_i \in D|\Xi^j) - \mathbb{P}(\Xi_i \in D)\right| + \right. \\
&\qquad\qquad \left. \sup_{D \subset \mathcal{D}} \left|\mathbb{P}(\Xi_i \in D) - \mathbb{P}(\Xi_i \in D|\Xi^{j-1})\right|\right\} \\
&\le 2\|f\|_\infty \left(1 + 2\sum_{i=j+1}^{n} g(i-j) + g(n-j+1)\right) \qquad \text{(Assumption 1)} \\
&\le 2G^* \|f\|_\infty. \qquad\qquad\qquad\qquad\qquad\qquad\qquad\qquad\qquad\qquad (20)
\end{aligned}
$$

Here $g(h)$ is given in Definition 17 and we exploited the fact that $\sqrt{g(h)} \le 1$ in the last inequality. This concludes the proof owing to the Azuma inequality for rectangular matrices (Remark 7.3, Tropp (2012)). $\square$

**Proposition 26** (Bernstein type). *Let $f$ be a $\mathbb{R}$-valued function on $\mathcal{D}$ and $Y := \sum_{i=1}^{n} f(\Xi_i)$. Then, under Assumption 1,*

$$
|Y - \mathbb{E}[Y]| \le \sqrt{2(\mathbb{V}[Y] + C_1)\ln(4/\delta)} + 4G^* \|f\|_\infty \ln(4/\delta)
$$

*with probability $1 - \delta$, where $\|f\|_\infty := \text{ess sup}_{\xi \in \mathcal{D}} |f(\xi)|$ and $C_1 := 4(G^* - 1)^2 \|f\|_\infty^2 \sqrt{2n \ln(2/\delta)}$.*

*Accordingly,*

$$|Y - \mathbb{E}[Y]| \leq \sqrt{2\mathbb{V}[Z]\ln(4/\delta)} + \mathcal{O}\left(G^* \|f\|_\infty n^{1/4} \ln^{3/4}(2/\delta)\right),$$

*if* $\ln(2/\delta) = \mathcal{O}(n)$.

*Proof.* Let $\mathbb{E}_j[\cdot]$ and $\mathbb{V}_j[\cdot]$ denote $\mathbb{E}[\cdot|\Xi^j]$ and $\mathbb{V}[\cdot|\Xi^j]$, $0 \leq j \leq n$, respectively. Let $Y_j := \mathbb{E}_j[Y]$ and $W_j := \sum_{i=1}^{j} \mathbb{V}_{i-1}[Y_i]$, $0 \leq j \leq n$. Note $|Y_j - Y_{j-1}| \leq L := 2G^*\|f\|_\infty$ by (20).

Freedman's inequality (Theorem 1.6, Freedman (1975)) states that

$$\mathbb{P}\left(|Y_n - Y_0| \geq a \ \wedge \ W_n \leq b\right) \leq 2\exp\left(-\frac{a^2}{2(La + b)}\right)$$

for all $a, b > 0$. Accordingly,

$$\mathbb{P}(W_n \leq b) \geq 1 - \delta/2 \quad \Rightarrow \quad \mathbb{P}(|Y_n - Y_0| \leq a^*) \geq 1 - \delta, \tag{21}$$

where $a^* := \sqrt{2b\ln(4/\delta)} + 2L\ln(4/\delta)$.

To show the concentration of $W_n$, consider the Doob martingale given by $Z_j := \mathbb{E}_j[W_n]$, $0 \leq j \leq n$. Let $f_i := f(\Xi_i)$, $1 \leq i \leq n$. Then, the difference sequence of $\{Z_j\}$ is bounded by

$$
\begin{aligned}
|Z_j - Z_{j-1}| &= |(\mathbb{E}_j - \mathbb{E}_{j-1})[W_n]| \\
&= \left|(\mathbb{E}_j - \mathbb{E}_{j-1})\sum_{i=1}^{n}\mathbb{V}_{i-1}[Y_i]\right| && \text{(Definition of } W_n) \\
&= \left|(\mathbb{E}_j - \mathbb{E}_{j-1})\sum_{i=j+1}^{n}\mathbb{V}_{i-1}[Y_i]\right| \\
&= |(\mathbb{E}_j - \mathbb{E}_{j-1})\mathbb{V}_j[Y]| && \text{(Law of total variance)} \\
&= \left|\sum_{j+1 \leq i,\ell \leq n}(\mathbb{E}_j - \mathbb{E}_{j-1})[(f_i - \mathbb{E}_j f_i)(f_\ell - \mathbb{E}_j f_\ell)]\right|
\end{aligned}
$$

Note that

$$
\begin{aligned}
&(\mathbb{E}_j - \mathbb{E}_{j-1})[(f_i - \mathbb{E}_j f_i)(f_\ell - \mathbb{E}_j f_\ell)] \\
&\leq (\mathbb{E}_j - \mathbb{E}_{j-1})\left[\theta(f_i - \mathbb{E}_j f_i)^2 + \frac{1}{\theta}(f_\ell - \mathbb{E}_j f_\ell)\right] && \text{(AM-GM inequality)} \\
&\leq 8\|f\|_\infty^2\left\{\theta g(i-j) + \frac{1}{\theta}g(\ell-j)\right\} && \text{(Definition 17)} \\
&\leq 16\|f\|_\infty^2\sqrt{g(i-j)g(\ell-j)} && (\theta = \sqrt{g(\ell-j)/g(i-j)})
\end{aligned}
$$

and therefore

$$
\begin{aligned}
|Z_j - Z_{j-1}| &\leq 16\|f\|_\infty^2 \sum_{j+1 \leq i,\ell \leq n}\sqrt{g(i-j)g(\ell-j)} \\
&\leq 16\|f\|_\infty^2\left(\frac{G^*-1}{2}\right)^2 && \text{(Assumption 1)} \\
&\leq 4\|f\|_\infty^2(G^*-1)^2
\end{aligned}
$$

Finally, the Azuma inequality yields

$$W_n - \mathbb{V}[Y] = Z_n - Z_0 \leq 4(G^*-1)^2\|f\|_\infty^2\sqrt{2n\ln(2/\delta)},$$

with probability $1 - \delta/2$. Combining it with (21),

$$|Y - \mathbb{E}[Y]| \leq \sqrt{2(\mathbb{V}[Y] + C_1)\ln(4/\delta)} + 4G^*\|f\|_\infty\ln(4/\delta)$$

with probability $1 - \delta$. $\qquad\square$

# F  Proofs of Propositions in the Main Text

## F.1  Proposition 1

*Proof.* The first argument is straightforward since all the pairs $(\Xi_i, \Xi_j)$ with $|i - j| \geq H$ are statistically mutually independent.

Let $d(h)$ and $\bar{d}(h)$ be defined as in Section 4.4 of Levin and Peres (2017). The second argument follows from the inequality

$$
\begin{aligned}
g(h) &\leq \bar{d}(h)\\
&\leq \bar{d}(t_{\text{mix}})^{\lfloor h/t_{\text{mix}}\rfloor}\\
&\leq \{2d(t_{\text{mix}})\}^{\lfloor h/t_{\text{mix}}\rfloor}\\
&\leq 2^{-\lfloor h/t_{\text{mix}}\rfloor}
\end{aligned}
$$

for all $h \geq 1$, where the first inequality is owing to the definition of $g(h)$ and $\bar{d}(h)$, the second and the third inequalities are shown by Lemma 4.12 and 4.11 of Levin and Peres (2017), respectively, and the last inequality is owing to the definition of the mixing time. $\qquad\square$

## F.2  Proposition 2

*Proof.* Let $(b^*, F^*)$ be the least-norm parameter that satisfy (2), where the norms are measured in the $\ell_2^K$- and the Frobenius metric. In the following, we prove the general case where the concentrability (Assumption 3) does not necessarily hold, $J(\pi) = b^{*\top}(I - \gamma F^*)x_0$. Then Proposition 2 immediately follows since the concentrability implies the uniqueness of the solution of (2), $(b^*, F^*) = (b, F)$.

Let $V_0 \in \mathbb{R}^{K \times K_0}$, $K_0 \leq K$, be a semi-orthogonal matrix such that $\text{span}\, V_0 = \text{span}\, \phi(\mathcal{S} \times \mathcal{A})$, where $\text{span}\, X$ denotes the linear span of a vector set $X$ and $V_0$ is interpreted as a $K'$-set of column vectors. Let $W_0 \in \mathbb{R}^{K \times (K - K_0)}$ be the complement of $V_0$, i.e., $[V_0, W_0] \in \mathbb{R}^{K \times K}$ is an orthogonal matrix. Note that $V_0 V_0^\top \phi = \phi$ and $W_0 \phi \equiv 0$ by definition.

Since $b^*$ and $F^*$ are the least-norm solutions, we have $b^* = V_0 \tilde{b}$ and $F^* = V_0 \tilde{F} V_0$ for $\tilde{b} := V_0^\top b^*$ and $\tilde{F} := V_0^\top F^* V_0$. Similarly, we have $x_0 = V_0 \tilde{x}_0$ for $\tilde{x}_0 := V_0^\top x_0$ since $x_0 = \langle \phi \rangle_{p_0^\pi} \in \text{span}\, V_0$.

Let $x_h := \langle P^h \phi \rangle_{p_0^\pi}$, i.e., the expected feature after $h$ transitions starting from $p_0$. Then, we can see $x_h := F^{*h} x_0$ from the recursion with the linear equations (2), and therefore $\langle P^h \bar{r} \rangle_{p_0^\pi} = b^{*\top} x_h = b^{*\top} F^{*h} x_0 = \tilde{b}^\top \tilde{F}^h \tilde{x}_0$. Substituting this to (1) yields the desired result,

$$
\begin{aligned}
J(\pi) &= \tilde{b}^\top \left( \sum_{h=0}^{\infty} \gamma^h \tilde{F}^h \right) \tilde{x}_0\\
&= \tilde{b}^\top (I - \gamma \tilde{F})^{-1} \tilde{x}_0 && \text{(if the sum converges)}\\
&= \begin{bmatrix} \tilde{b} \\ 0 \end{bmatrix}^\top \begin{bmatrix} V_0^\top \\ W_0^\top \end{bmatrix} [V_0, W_0] \begin{bmatrix} I - \gamma \tilde{F} & 0 \\ 0 & I \end{bmatrix}^{-1} \begin{bmatrix} V_0^\top \\ W_0^\top \end{bmatrix} [V_0, W_0] \begin{bmatrix} \tilde{x}_0 \\ 0 \end{bmatrix} && \text{(}[V_0, W_0] \text{ is orthogonal)}\\
&= b^{*\top} \left\{ [V_0, W_0] \begin{bmatrix} I - \gamma \tilde{F} & 0 \\ 0 & I \end{bmatrix} \begin{bmatrix} V_0^\top \\ W_0^\top \end{bmatrix} \right\}^{-1} x_0\\
&= b^{*\top} (I - \gamma F^*)^{-1} x_0,
\end{aligned}
$$

given $\sum_{h=0}^{\infty} \gamma^h \tilde{F}^h$ converges.

The convergence of the infinite sum is shown as follows. Note that

$$
\begin{aligned}
\|\tilde{F}^h V_0^\top \phi(s, a)\|_2 &= \|V_0 \tilde{F}^h V_0^\top \phi(s, a)\|_2\\
&= \|F^{*h} \phi(s, a)\|_2\\
&= \|\langle P^h \phi \rangle_{p_0^\pi}\|_2\\
&\leq \langle \|P^h \phi\|_2 \rangle_{p_0^\pi} \leq 1, && \text{(Assumption 2)}
\end{aligned}
$$

for all $s, a \in \mathbb{R}^K$ and arbitrarily large $h \geq 0$. Thus, since $V_0^\top \phi$ is full rank, i.e., span $V_0^\top \phi(\mathcal{S} \times \mathcal{A}) = \mathbb{R}^{K_0}$, the spectral radius of $\tilde{F}$ is no larger than 1. This concludes the proof. $\qquad\square$

### F.3 Proposition 3

*Proof.* Observe

$$
\begin{aligned}
& \mathbb{E}\, \mathcal{C}(b, F; \Xi^n) \\
&= \frac{1}{n} \sum_{i=1}^n \mathbb{E}\left[\left|r_i - b^\top \phi(s_i, a_i)\right|^2 + |\psi_\pi(s_i') - F\phi(s_i, a_i)|^2\right] \\
&= \frac{1}{n} \sum_{i=1}^n \mathbb{E}\mathbb{E}\left[\left|\bar{r}(s_i, a_i) - b^\top \phi(s_i, a_i)\right|^2 + |(P\phi)(s_i, a_i) - F\phi(s_i, a_i)|^2 \,\Big|\, s_i, a_i\right] + \\
& \quad \frac{1}{n} \sum_{i=1}^n \left\{\mathbb{V}\left[r_i \mid s_i, a_i\right] + \mathbb{V}\left[\psi_\pi(s_i') \mid s_i, a_i\right]\right\} \\
&= D^2(b, F) + \frac{1}{n} \sum_{i=1}^n \left\{\mathbb{V}\left[r_i \mid s_i, a_i\right] + \mathbb{V}\left[\psi_\pi(s_i') \mid s_i, a_i\right]\right\},
\end{aligned}
$$

which yields the proposition since the second term is a constant with respect to $(b, F)$. $\qquad\square$

### F.4 Proposition 4

*Proof.* Note that $(\hat{b}, \hat{F})$ is the least-norm solution to the stationary-point equation $\nabla_{b,F}\mathcal{C}(b, F; \xi_n) = 0$. More concretely,

$$
\begin{aligned}
\nabla_b \mathcal{C}(b, F; \xi_n) = 0 &\Leftrightarrow \Phi^\top(\Phi b - \hat{r}) = 0, \\
\nabla_F \mathcal{C}(b, F; \xi_n) = 0 &\Leftrightarrow \Phi^\top(\Phi F^\top - \Psi_\pi) = 0.
\end{aligned}
$$

It is straightforward to check if the explicit formulae in the proposition satisfy the above equations and have the least norms. $\qquad\square$

### F.5 Proposition 5

*Proof.* Observe the vector $\theta_h \in \mathbb{R}^K$ of the $h$-th iteration of Algorithm 2 is given by

$$
\theta_h = \frac{1}{n}\hat{\Sigma}^+ \Phi^\top(\hat{r} + \gamma \Psi_\pi \theta_{h-1}) = \hat{b} + \gamma \hat{F}^\top \theta_{h-1}.
$$

Suppose $\theta_h$ converges to $\theta_*$. Then we have $\theta_* = \hat{b} + \gamma\hat{F}^\top\theta_*$, which implies $\theta_* = (I - \gamma\hat{F}^\top)\hat{b} = \hat{F}_\gamma^\top \hat{b}$. This completes the proof. $\qquad\square$

### F.6 Proposition 7

*Proof.* We first show $Q^\pi \in \mathcal{F}_\phi$ if $\mathcal{R}_B = 0$. Observe that $\mathcal{R}_B = 0$ implies $BQ^\sharp = Q^\sharp$, which is exactly the Bellman equation and the solution is unique, $Q^\sharp = Q^\pi$. This implies $Q^\pi \in \mathcal{F}_\phi$ since $Q^\sharp \in \mathcal{F}_\phi$.

We now turn to the inverse direction, i.e., to show $\mathcal{R}_B = 0$ if $Q^\pi \in \mathcal{F}_\phi$. Let $\theta^\pi, \theta^\sharp \in \mathbb{R}^K$ be the least-norm coefficients of $Q^\pi$ and $Q^\sharp$, respectively, i.e., $Q^\pi(s, a) = \theta^{\pi\top}\phi(s, a)$ and $Q^\sharp(s, a) = \theta^{\sharp\top}\phi(s, a)$ for $s \in \mathcal{S}$ and $a \in \mathcal{A}$. The true value function is the unique solution of the Bellman equation,

$$
Q^\pi = BQ^\pi = \bar{r} + \gamma PQ^\pi,
$$

which implies

$$
\phi^\top\theta^\pi = \bar{r} + \gamma P(\phi^\top\theta^\pi).
$$

Multiplying $\phi$ and taking expectation with respect to the data marginal, we get

$$\left\langle \phi\phi^\top \right\rangle_\mu \theta^\pi = \left\langle \bar{r}\phi + \gamma\phi P(\phi^\top \theta^\pi) \right\rangle_\mu$$
$$= \left\langle \bar{r}\phi \right\rangle_\mu + \gamma \left\langle \phi\psi_\pi^\top \right\rangle_\mu \theta^\pi.$$

Note that $b^\sharp = \Sigma^+ \left\langle \bar{r}\phi \right\rangle_\mu$, $F^\sharp = \left\langle \phi\psi_\pi^\top \right\rangle_\mu \Sigma^+$ and $\Sigma = \left\langle \phi\phi^\top \right\rangle_\mu$ by the definition of $(b^\sharp, F^\sharp)$. Therefore, $\theta^\pi$ is a solution of the following equation,

$$\Sigma^+\Sigma\theta^\pi = b^\sharp + \gamma F^{\sharp\top}\theta^\pi,$$

which is uniquely solved with $\theta^\pi = (I - \gamma F^{\sharp\top})^{-1}b^\sharp = \theta^\sharp$ under the concentrability condition. Therefore $Q^\sharp = Q^\pi$. Finally, the Bellman equation yields the desired result,

$$\mathcal{R}_B = BQ^\sharp - Q^\sharp = BQ^\pi - Q^\pi = 0.$$

$\square$

### F.7   Proposition 9

*Proof.* Let $\mathcal{U}$ be a linear mapping from $K$-vectors to functions over $\mathcal{S} \times \mathcal{A}$ such that $\mathcal{U}x = x^\top\phi$, $x \in \mathbb{R}^K$. Also, let $\Pi$ be a linear mapping of functions on $\mathcal{S} \times \mathcal{A}$ such that $\Pi f = \mathcal{U}\Sigma^+ \left\langle f\phi \right\rangle_\mu$, $f : \mathcal{S} \times \mathcal{A} \to \mathbb{R}$.

Now we have $\Pi$ is non-expansive since, for all $s \in \mathcal{S}$ and $a \in \mathcal{A}$,

$$
\begin{aligned}
|(\Pi f)(s, a)| &= \left|\left\langle f\phi^\top \right\rangle_\mu \Sigma^+\phi(s, a)\right| \\
&\leq \left\langle \left|f\phi^\top\Sigma^+\phi(s, a)\right| \right\rangle_\mu && \text{(Jensen's)} \\
&\leq \|f\|_\infty \left\langle \phi^\top\Sigma^+\phi(s, a) \right\rangle_\mu && \text{(Nonnegativity of } \phi(s', a')^\top\Sigma^+\phi(s, a)) \\
&= \|f\|_\infty \left\langle \phi \right\rangle_\mu^\top \Sigma^+\phi(s, a) \\
&= \|f\|_\infty v_0^\top \phi(s, a) \\
&= \|f\|_\infty,
\end{aligned}
$$

where the second last equality follows from $\Sigma v_0 = \left\langle \phi\phi^\top \right\rangle_\mu v_0 = \left\langle (v_0^\top\phi)\phi \right\rangle_\mu = \left\langle \phi \right\rangle_\mu$ and the invertibility of $\Sigma$.

Moreover, we have $\mathcal{U}F^{\sharp\top} = \Pi P\mathcal{U}$ since, for all $x \in \mathbb{R}^K$,

$$
\begin{aligned}
\mathcal{U}F^{\sharp\top}x &= \mathcal{U}\Sigma^{-1}\left\langle \phi(P\phi)^\top \right\rangle_\mu x && \text{(Corollary 22)} \\
&= \mathcal{U}\Sigma^{-1}\left\langle \phi(P(x^\top\phi)) \right\rangle_\mu \\
&= \mathcal{U}\Sigma^{-1}\left\langle \phi(P\mathcal{U}x)) \right\rangle_\mu \\
&= \Pi(P\mathcal{U}x).
\end{aligned}
$$

Thus, $\mathcal{U}(F^{\sharp\top})^h$ is non-expansive for all $h \geq 1$ since $\mathcal{U}(F^{\sharp\top})^h = \Pi P\mathcal{U}(F^{\sharp\top})^{h-1} = ... = (\Pi P)^h\mathcal{U}$ and $\Pi$, $P$ and $\mathcal{U}$ are all non-expansive. This implies

$$
\begin{aligned}
\left|x^\top F^{\sharp h}\phi(s, a)\right| &= \left|(\mathcal{U}F^{\sharp\top h}x)(s, a)\right| \\
&\leq \left\|\mathcal{U}(F^{\sharp\top})^h x\right\|_\infty \\
&\leq \|x\|_2
\end{aligned}
$$

for all $h \geq 1$, $x \in \mathbb{R}^K$, $s \in \mathcal{S}$ and $a \in \mathcal{A}$, which implies $\left\|F^{\sharp h}\phi(s, a)\right\|_2 \leq 1$ for all $h \geq 1$, $s \in \mathcal{S}$ and $a \in \mathcal{A}$. Thus, we have $\lambda_1(F^\sharp) \leq 1$ and $F_\gamma^\sharp = (I - \gamma F^\sharp)^{-1}$ exists.

Observe

$$
\begin{aligned}
\|F_\gamma^\sharp\|_2^2 &= \operatorname{tr}\left[F_\gamma^\sharp F_\gamma^{\sharp\top}\right] \\
&\le \operatorname{tr}\left[F_\gamma^\sharp\left(c_*^{-1}\Sigma + WW^\top\right)F_\gamma^{\sharp\top}\right] && (c_*^{-1}\Sigma + WW^\top \succeq I) \\
&= \frac{1}{c_*}\operatorname{tr}\left[F_\gamma^\sharp\Sigma F_\gamma^{\sharp\top}\right] + (K - \tilde{K}) && (F_\gamma^\sharp W = W) \\
&= \frac{1}{c_*}\left\langle\|F_\gamma^\sharp\phi\|_2^2\right\rangle_\mu + (K - \tilde{K}) \\
&\le \frac{1}{(1-\gamma)^2 c_*} + (K - \tilde{K}),
\end{aligned}
$$

where the RHS is independent of $\pi$. Here, the last inequality follows from

$$
\begin{aligned}
\left\|F_\gamma^\sharp\phi(s,a)\right\|_2 &= \left\|\sum_{h\ge 0}\gamma^h F^{\sharp h}\phi(s,a)\right\|_2 && (\lambda_1(F^\sharp) \le 1) \\
&\le \sum_{h\ge 0}\gamma^h\left\|F^{\sharp h}\phi(s,a)\right\|_2 \\
&\le \frac{1}{1-\gamma}. && \left(\left\|F^{\sharp h}\phi(s,a)\right\|_2 \le 1\right) \quad\quad (22)
\end{aligned}
$$

$\square$

### F.8 Proposition 10

The proposition is a special case of the following result.

**Definition 18** (Refining tabular sequence)**.** *We call $\phi$ as a refining tabular sequence if, for all $m \ge 1$, $\phi^{(m)}$ is tabular with respect to an $K_m$-partition $\{\mathcal{P}_k^{(m)}\}_{k=1}^{K_m}$ of $\mathcal{S} \times \mathcal{A}$ such that $\lim_{m\to\infty}\min_{k\in[m]} K_m\operatorname{vol}(\mathcal{P}_k^{(m)}) > 0$ and $\lim_{m\to\infty}\max_{k\in[m]}\operatorname{diam}(\mathcal{P}_k^{(m)}) = 0$.*

**Proposition 27.** *$(\phi, \hat{m})$ is consistent for all refining tabular sequences $\phi$ if*

1. *$\Xi^n$ is $G^*$-mixing for some $G^* < \infty$ (Assumption 1).*

2. *$0 < \inf_{s\in\mathcal{S}, a\in\mathcal{A}}\mu(s,a)$ and $\sup_{s\in\mathcal{S}, a\in\mathcal{A}}\mu(s,a) < \infty$.*

3. *$K_{\hat{m}(n)}/\sqrt{n} \to 0$ as $n \to \infty$.*

*Proof.* Let $\mathcal{R}_B^{(m)}$ and $\mathcal{R}_\chi^{(m)}$ be the residual functions for $\phi^{(m)}$. First, we show $\Delta_m := \hat{J}(\pi; \phi^{(m)}) - J(\pi) + \langle\mathcal{R}_B^{(m)}\mathcal{R}_\chi^{(m)}\rangle_\mu/(1-\gamma)$ converges to zero as $n \to \infty$, where $m = \hat{m}(n)$. Then, second, we show $\langle\mathcal{R}_B^{(m)}\mathcal{R}_\chi^{(m)}\rangle_\mu \to 0$ as $m \to \infty$.

For the first step, we examine the asymptotic behavior of the unbiased term $\Delta_{\hat{m}(n)}$. According to Theorem 13 and Proposition 14, we have

$$
\Delta_m = \mathcal{O}\left(\frac{\bar{V}^{(m)}\rho^{(m)}G^{*\frac{1}{2}}\ln^{\frac{1}{2}}\frac{2}{\delta}}{c_*^{(m)\frac{1}{2}}n^{\frac{1}{2}}} + \frac{\bar{V}^{(m)}\rho^{(m)}G^*\ln^{\frac{3}{4}}\frac{2}{\delta}}{c_*^{(m)}n^{\frac{3}{4}}} + \frac{\bar{V}^{(m)}\rho^{(m)2}G^{*2}\ln\frac{2K}{\delta}}{c_*^{(m)2}n}\right),
$$

where $\bar{V}^{(m)}$, $\rho^{(m)}$ and $c_*^{(m)}$ are defined associated with each $\phi^{(m)}$. An elementary calculation shows $\bar{V}^{(m)} \le 1 + 2/(1-\gamma)$ and $\rho^{(m)} \le 1/(1-\gamma)$ for all tabular $\phi^{(m)}$. Thus, the only $m$-dependent factor is $c_*^{(m)}$, lower bounded by

$$
\begin{aligned}
c_*^{(m)} &= \min_{k\in[K_m]}\int_{\mathcal{P}_k^{(m)}}\mu(s,a)\,\mathrm{d}s\,\mathrm{d}a \\
&\ge \inf_{s\in\mathcal{S}, a\in\mathcal{A}}\mu(s,a)\min_{k\in[K_m]}\operatorname{vol}(\mathcal{P}_k^{(m)}) \\
&\ge \frac{C_2}{K_m}\inf_{s\in\mathcal{S}, a\in\mathcal{A}}\mu(s,a) && (\text{Definition 18})
\end{aligned}
$$

for sufficiently large $m$. Hiding the dependencies on the variables other than $n$ and $m$, we get

$$\Delta_m = \mathcal{O}\left(\frac{K_m^{1/2}}{n^{1/2}} + \frac{K_m}{n^{3/4}} + \frac{K_m^2}{n}\right).$$

This implies $\Delta_{\hat{m}(n)} \to 0$ since $K_m/\sqrt{n} \to 0$.

For the second step, we utilize Hölder's inequality. Let $f^{(m)} := \operatorname{argmin}_{f \in \mathcal{F}_{\phi(m)}} \|\frac{\nu}{\mu} - f\|_{\mu,1}$, where $\|\cdot\|_{\mu,1}$ denotes the $L^1(\mu)$-norm. Then

$$\langle \mathcal{R}_B^{(m)} \mathcal{R}_\chi^{(m)} \rangle_\mu = \left\langle \mathcal{R}_B^{(m)} \left(\frac{\nu}{\mu} - f^{(m)}\right) \right\rangle_\mu \qquad\qquad (\mathcal{R}_B^{(m)} \in \mathcal{F}_{\phi(m)})$$

$$\leq \left\| \mathcal{R}_B^{(m)} \right\|_\infty \left\| \frac{\nu}{\mu} - f^{(m)} \right\|_{\mu,1}$$

$$\leq \left(1 + \frac{2}{1-\gamma}\right) \left\| \frac{\nu}{\mu} - f^{(m)} \right\|_{\mu,1},$$

where in the last inequality, we exploit $\left\| \mathcal{R}_B^{(m)} \right\|_\infty \leq \bar{V}^{(m)} \leq 1 + 2/(1-\gamma)$. The second step is completed by showing $\|\frac{\nu}{\mu} - f^{(m)}\|_{\mu,1} \to 0$. Fix arbitrary $\varepsilon > 0$. Note that the set of continuous functions on $T := \mathcal{S} \times \mathcal{A}$ is dense in $L^1(T)$ since $T$ is a compact subset of a product of Euclidean spaces and discrete spaces. This implies there exists a continuous function $\hat{f}$ such that $\|\frac{\nu}{\mu} - \hat{f}\|_1 < \varepsilon$. Every continuous function on $T$ is uniformly continuous since it is compact. Every uniformly continuous function is approximated in the uniform norm with some piecewise constant function $f^{(m)*} \in \mathcal{F}_{\phi(m)}$ arbitrarily well as $m \to \infty$ since the largest diameter of the constant cells $\mathcal{P}_k^{(m)}$, $k \in [K_m]$, is shrinking. This implies there exists $m_0 \geq 1$ such that $\|\hat{f} - f^{(m)*}\|_\infty < \varepsilon$ if $m \geq m_0$. Finally, we get

$$\left\| \frac{\nu}{\mu} - f^{(m)} \right\|_{\mu,1} \leq \left\| \frac{\nu}{\mu} - f^{*(m)} \right\|_{\mu,1} \qquad\qquad (f^{(m)} \text{ is the minimizer})$$

$$\leq \left\| \frac{\nu}{\mu} - \hat{f} \right\|_{\mu,1} + \left\| \hat{f} - f^{*(m)} \right\|_{\mu,1}$$

$$\leq C_\mu \left\| \frac{\nu}{\mu} - \hat{f} \right\|_1 + \left\| \hat{f} - f^{*(m)} \right\|_\infty$$

$$\leq (1 + C_\mu)\varepsilon$$

for $m \geq m_0$. $\qquad\square$

### F.9  Proposition 11

The proposition is a special case of the following result.

**Proposition 28.** *Let*

$$d^* := 1 \vee \limsup_{m \to \infty} \frac{\ln K_m}{\min_{k \in [K_m]} \ln \frac{1}{\operatorname{diam}(\mathcal{P}_k^{(m)})}}.$$

*Under the assumptions of Proposition 27, if $\nu/\mu$ are Lipschitz continuous, then we have $|\hat{J}(\pi; \phi, \hat{m}) - J(\pi)| = \mathcal{O}(n^{-\frac{1-\varepsilon}{2d^*+1}})$ for all $\varepsilon > 0$ with taking $\hat{m}(n)$ such that $K_{\hat{m}(n)} = \Theta(n^{\frac{d^*}{2d^*+1}})$.*

*Proof.* Without loss of generality, we assume $\nu/\mu$ is 1-Lipschitz continuous. The strategy is similar to Proposition 27. The error is decomposed as $\hat{J}(\pi; \phi^{(m)}) - J(\pi) = -\frac{1}{1-\gamma}\langle \mathcal{R}_B^{(m)} \mathcal{R}_\chi^{(m)} \rangle_\mu + \Delta_m$. The convergence rate of the unbiased term $\Delta_m$, $m = \hat{m}(n)$, is established as in the proof of Proposition 27,

$$\Delta_m = \mathcal{O}\left(\frac{K_m^{1/2}}{n^{1/2}} + \frac{K_m}{n^{3/4}} + \frac{K_m^2}{n}\right)$$

Taking $m = \hat{m}(n)$, the right-hand side is evaluated as $\mathcal{O}(n^{-\frac{1}{2d^*+1}})$ since

$$\frac{K_{\hat{m}(n)}^{1/2}}{n^{1/2}} + \frac{K_{\hat{m}(n)}}{n^{3/4}} + \frac{K_{\hat{m}(n)}^2}{n} = n^{\frac{d^*}{2(2d^*+1)} - \frac{1}{2}} + n^{\frac{d^*}{2d^*+1} - \frac{3}{4}} + n^{\frac{2d^*}{2d^*+1} - 1}$$

$$= n^{-\frac{1}{2d^*+1}\frac{d^*+1}{2}} + n^{-\frac{1}{2d^*+1}\frac{2d^*+3}{4}} + n^{-\frac{1}{2d^*+1}}$$

$$= \mathcal{O}(n^{-\frac{1}{2d^*+1}}). \hspace{3cm} (d^* \geq 1)$$

For the bias term, we show $\|\frac{\nu}{\mu} - f^{(m)}\|_{\mu,1} = \mathcal{O}(K_m^{-(1-\varepsilon)/d^*})$ for all $\varepsilon > 0$, where $\|\cdot\|_{\mu,1}$ denotes the $L^1(\mu)$-norm and $f^{(m)} := \operatorname{argmin}_{f \in \mathcal{F}_{\phi^{(m)}}} \|\frac{\nu}{\mu} - f\|_{\mu,1}$. It then yields the desired rate by the same calculation as the proof of Proposition 27,

$$\langle \mathcal{R}_B^{(\hat{m}(n))} \mathcal{R}_\chi^{(\hat{m}(n))} \rangle_\mu \leq \left(1 + \frac{2}{1-\gamma}\right) \left\|\frac{\nu}{\mu} - f^{(\hat{m}(n))}\right\|_{\mu,1}$$

$$= \mathcal{O}(K_{\hat{m}(n)}^{-(1-\varepsilon)/d^*})$$

$$= \mathcal{O}(n^{-\frac{1-\varepsilon}{2d^*+1}}).$$

To bound $\|\frac{\nu}{\mu} - f^{(m)}\|_{\mu,1}$, recall by its definition that $f^{(m)}$ is the best approximation of $\nu/\mu$ in $\mathcal{F}_{\phi^{(m)}}$ with respect to the $L^1(\mu)$-norm. Since $\nu/\mu$ is 1-Lipschitz continuous and $\mathcal{F}_{\phi^{(m)}}$ contains any function having a constant value on each cell of the partition $\{\mathcal{P}_k^{(m)}\}_{k \in [K_m]}$, we have, for sufficiently large $m$,

$$\left\|\frac{\nu}{\mu} - f^{(m)}\right\|_{\mu,1} \leq \left\|\frac{\nu}{\mu} - f^{(m)}\right\|_\infty$$

$$\leq \max_{k \in [K_m]} \operatorname{diam}(\mathcal{P}_k^{(m)}).$$

$$= \exp\left(- \min_{k \in [K_m]} \ln \frac{1}{\operatorname{diam}(\mathcal{P}_k^{(m)})}\right).$$

$$\leq \exp\left(-\frac{1-\varepsilon}{d^*} \ln K_m\right) \hspace{2cm} \text{(Definition of } d^*\text{)}$$

$$= K_m^{-(1-\varepsilon)/d^*}.$$

$\square$