# OpenReview forum: "Asymptotically Exact Error Characterization of Offline Policy Evaluation with Misspecified Linear Models"
_NeurIPS.cc/2021/Conference — NeurIPS 2021 Poster_

### Official Review · Reviewer_S51y · 2021-07-10

**Rating:** 5
**Confidence:** 3

**Summary:**

In this paper, the authors analyze the off-policy evaluation error of a direct estimator with linear function approximation under unrealizability. In Theorem 6, they derive the closed form of the asymptotic bias term as the number of samples approaching \infty. Besides, in section 3.2, they study another method where the feature mappings can be selected adaptively and provide the convergence analysis for their estimation to ground-truth J(\pi).

**Limitations And Societal Impact:**

I think the authors adequately addressed the limitations and potential negative societal impact of their work.

**Main Review:**

This paper is well-written, and the notation and definitions are clear. Their theoretical results are solid and interesting. But I’m concerned about the significance of the contribution of this paper.

(1) I’m feeling the theoretical results in Theorem 6 may not be novel enough. The bias decomposition in Theorem 6 is similar to the Eq (12) in [1], in which the authors proposed a “doubly robust” estimator and decomposed the bias to the residual of density ratio estimation error and value estimation error.

It seems that if we use linear functions to approximate density ratio and value function (e.g. see Example 2 and Example 7 in [2]) with the same data and plug them into the “doubly-robust estimator” in [1], it directly recovers the \hat J(\pi) estimator in Algorithm 1 in this paper. As a result, a bias term similar to Theorem 6 can be obtained by directly adapting from results in [1] to the linear setting as a special case with a different definition of “residual” (R_\chi & R_B in this paper v.s. \epsilon_{\hat w} & \epsilon_{\hat V} in [1]).

I wonder if the authors can discuss some connections and essential differences comparing with the results in [1], which is missing in the paper.

(2) Although the analysis in this paper fills some blanks in OPE research, I'm feeling the results in this paper is not substantial enough. It would be a more solid paper to me if there are more discussions about what we can do with the bias analysis, or say, how can we use that bias analysis (e.g. given two feature sets, can we identify the one with a smaller bias), or maybe more bias analysis beyond the linear setting with unrealizability.

Reference:

[1] Doubly Robust Bias Reduction in Infinite Horizon Off-Policy Estimation, Ziyang Tang, Yihao Feng, Lihong Li, Dengyong Zhou, and Qiang Liu

[2] Minimax Weight and Q-Function Learning for Off-Policy Evaluation, Masatoshi Uehara, Jiawei Huang, and Nan Jiang

**Time Spent Reviewing:**

6

---

> ### Author Response · Authors · 2021-08-06
> **On Connections to DR Estimators**
>
> We thank Reviewer S51y for the valuable comments and suggestions.
>
> > It seems that if we use linear functions to approximate density ratio and value function (e.g. see Example 2 and Example 7 in [2]) with the same data and plug them into the “doubly-robust estimator” in [1], it directly recovers the \hat J(\pi) estimator in Algorithm 1 in this paper.
>
> We are not sure that recovers Algorithm 1. In fact, Example 7 of [2] alone is equivalent to LSTDQ according to the authors, which is equivalent to Algorithm 1. Thus, Theorem 6 shows that Algorithm 1 is doubly robust without density-ratio estimators such as Example 2 of [2].
>
> > I wonder if the authors can discuss some connections and essential differences comparing with the results in [1], which is missing in the paper.
>
> Please see the separate post on this point at https://openreview.net/forum?id=_y2G1-i7L8&noteId=VPJ6VKy-7z (we separate it since the other reviewers also suggested the need for the discussion).

---

> > ### Comment · Area_Chair_45QX · 2021-08-12
> > **LSTDQ**
> >
> > Examples 2 and 7 in [2] are exactly the same algorithm when it comes to estimating $J(\pi)$, i.e., they both become LSTDQ evaluated at the initial state distribution; see last sentence in the paragraph after Example 7. Therefore, they essentially give different interpretations to LSTDQ and one could freely use either when analyzing the method. When either the value-function or the density ratio is linear in the features, the bias is upper bounded by the corresponding optimization objectives (L_w or L_q), which is always 0 in the linear case, so asymptotically there is no bias.

---

> > > ### Author Response · Authors · 2021-08-13
> > > **Thank You for Clarification**
> > >
> > > Yes, both of Example 2 and 7 in [2] are individually exactly the same as Algorithm 1 when it comes to $J(\pi)$. Also, they indirectly imply that linear DM is doubly robust in bias. However, the built-in DR property of linear DM was not featured or characterized asymptotically. In particular, note the significant difference in their DR implications;
> > > Theorem 2 and 5 in [2] with linear features imply "minimum of min-max" bound ,
> > >
> > > $$|\hat{J}(\pi)-J(\pi)|\le \min_w \max_q L_{\mathrm{w}}(w, q) \wedge \min_q \max_w L_{\mathrm{q}}(q, w) +o(1), $$
> > >
> > > whereas our Theorem 6 with Cauchy--Schwarz inequality implies "min-min" bound (by our Definition 8 and 9),
> > >
> > > $$|\hat{J}(\pi)-J(\pi)|\le \min_w \min_q ||\nu/\mu-w||\_{L^2(\mu)} || B Q^\sharp - q||\_{L^2(\mu)} +o(1),$$
> > >
> > > where $B$ denotes the Bellman operator, and the minimums and maximums w.r.t. $w$ and $q$ are taken over $\mathcal{F}_\phi$.
> > >
> > > Also, it seems nontrivial whether combining these two estimators with the DR formula (Eq. (14) in [2] or Eq. (9) in [1]) recovers Algorithm 1 and Theorem 6.

---

> > > > ### Comment · Area_Chair_45QX · 2021-08-23
> > > > **Right**
> > > >
> > > > Right. I think the key difference between the two bounds is not "min of min-max" versus "min-min" per se; you can probably get rid of the "max" in the first bound by bounding the $L_w$ and $L_q$ terms using the difference between $w$ and $\nu/\mu$ and the Bellman error of $q$ (probably by Holder or Cauchy-Schwarz). That said, even if this is possible, one would get $\min$ of two error terms, whereas your analysis gives the *multiplication* of two error terms. That seems a qualitative difference.
> > > >
> > > > (Also fyi, when you make significant updates to a comment, it would be better to post a new comment otherwise no notification email is generated.)
> > > >
> > > > Re "recovering Alg 1", I didn't mean to plug the two estimators in DR in [1] or [2]. Since the two estimators are themselves literally LSTDQ when using linear classes (and the equivalence holds on a finite sample), and your Line 196 indicates "Algorithm 1 is equivalent to LSTDQ", it seems that a direct logical consequence is that "the two estimators, when using linear classes, are literally equivalent to Algorithm 1". Did I misunderstand something here?

---

> > > > > ### Author Response · Authors · 2021-08-25
> > > > > **We Agree**
> > > > >
> > > > > > That said, even if this is possible, one would get min of two error terms, whereas your analysis gives the multiplication of two error terms. That seems a qualitative difference.
> > > > >
> > > > > Right. This seems more clear-cut.
> > > > >
> > > > > > (Also fyi, when you make significant updates to a comment, it would be better to post a new comment otherwise no notification email is generated.)
> > > > >
> > > > > Thank you for heads-up. Certainly we will do so.
> > > > >
> > > > >
> > > > >
> > > > > > Re "recovering Alg 1", I didn't mean to plug the two estimators in DR in [1] or [2]. Since the two estimators are themselves literally LSTDQ when using linear classes (and the equivalence holds on a finite sample), and your Line 196 indicates "Algorithm 1 is equivalent to LSTDQ", it seems that a direct logical consequence is that "the two estimators, when using linear classes, are literally equivalent to Algorithm 1". Did I misunderstand something here?
> > > > >
> > > > > That was not a direct response to you (appologies for confusion), but a reference to the original context of the thread, that Reviewer S51y’s mention on the novelty of Theorem 6 compared to [2]:
> > > > >
> > > > > > It seems that if we use linear functions to approximate density ratio and value function (e.g. see Example 2 and Example 7 in [2]) with the same data and plug them into the “doubly-robust estimator” in [1], it directly recovers the \hat J(\pi) estimator in Algorithm 1 in this paper.

---

> > > > > > ### Comment · Reviewer_S51y · 2021-09-02
> > > > > > **More Explanation about My First Point**
> > > > > >
> > > > > > Thanks for the authors' response, and I would like to explain more about "use Example 2 & 7 in [2] and plug it into [1] to recover the estimator in Algorithm 1".
> > > > > >
> > > > > > Suppose we use the same data to estimate the density ratio and the Q function with Example 2 and Example 7 in [2], what we have is the following two estimators for density ratio and Q functions.
> > > > > > $$
> > > > > > \hat w(\tilde{s},\tilde{a}) = (1-\gamma) E_{d_0}[\phi(s,\pi_e)^\top]\Sigma^{-1} \phi(\tilde{s},\tilde{a}) ,\quad \hat q(\tilde{s},\tilde{a})=\phi(\tilde{s},\tilde{a})^\top \Sigma^{-1} E_n[r\phi(s,a)]
> > > > > > $$
> > > > > > where for simplicity, I use $\Sigma$ as a short note of $E_n[-\gamma \phi(s,a)\phi(s',\pi_e)^\top+\phi(s,a)\phi(s,a)^\top]$.
> > > > > >
> > > > > > As a result, if we plug the above two estimators in the DR-estimator in [1] (and use the same dataset), we should have:
> > > > > > $$
> > > > > > v_{DR}=(1-\gamma)E_{s_0\sim d_0}[\hat q(s_0,\pi_e)]+E_n[\hat w(s,a)(r+\gamma \hat q(s',\pi_e)-\hat q(s,a))]
> > > > > > $$
> > > > > > Observe that for arbitrary $s_1,a_1,s_2,a_2$, we have:
> > > > > > $$
> > > > > > \hat w(s_1,a_1) \hat q(s_2,a_2) =  (1-\gamma) E_{d_0}[\phi(s,\pi_e)^\top] \Sigma^{-1}   \Big(\phi(s_1,a_1)\phi(s_2,a_2)^\top\Big) \Sigma^{-1} E_n[r\phi(s,a)]
> > > > > > $$
> > > > > > Because we use the same dataset to build the DR-estimator, we have
> > > > > > $$
> > > > > >  E_n[\hat w(s,a) (\gamma \hat q(s',\pi_e) - \hat q(s,a))] = (1-\gamma) E_{d_0}[\phi(s,\pi_e)^\top] \Sigma^{-1}   E_n\Big[\Big(\gamma \phi(s,a)\phi(s',\pi_e)^\top-\phi(s,a)\phi(s,a)^\top\Big)\Big] \Sigma^{-1} E_n[r\phi(s,a)] = -(1-\gamma) E_{d_0}[\phi(s,\pi_e)^\top] \Sigma^{-1}   E_n[r\phi(s,a)]
> > > > > > $$
> > > > > > which implies that
> > > > > > $$
> > > > > > E_n[\hat w(s,a) (r+\gamma \hat q(s',\pi_e) - \hat q(s,a))] = 0
> > > > > > $$
> > > > > > Therefore, the DR estimator will reduce to
> > > > > > $$
> > > > > > v_{DR} = (1-\gamma)E_{s_0\sim d_0}[\hat q(s_0,\pi_e)] = (1-\gamma)E_{s_0\sim d_0}[\phi(s_0,\pi_e)^\top] \Sigma^{-1} E_n[r\phi(s,a)]
> > > > > > $$
> > > > > > which is exactly the estimator in Algorithm 1 in the paper (or say the LSTDQ estimator).
> > > > > >
> > > > > > As a result, we can apply the techniques in [1] to directly obtain a bias analysis:
> > > > > > $$
> > > > > > v_{DR}-v^\pi =E_{d_0}[\epsilon_{\hat w}\epsilon_{\hat q}]
> > > > > > $$
> > > > > > where $\epsilon_{\hat w}$ and $\epsilon_{\hat q}$ are estimation error of $\hat w$ and $\hat q$.
> > > > > >
> > > > > >
> > > > > > Please let me know if I made any mistake. And if the above derivations are correct, I wonder if the authors can explain the connection and difference between their results and the bias analysis of DR-estimator derived in the way above, given that $\hat w$ and $\hat q$ are also obtained by minimizing some quadratic loss function and the bias is also the multiplication between the bias of w and q estimators?
> > > > > >
> > > > > >
> > > > > > Reference:
> > > > > >
> > > > > > [1] Doubly Robust Bias Reduction in Infinite Horizon Off-Policy Estimation, Ziyang Tang, Yihao Feng, Lihong Li, Dengyong Zhou, and Qiang Liu
> > > > > >
> > > > > > [2] Minimax Weight and Q-Function Learning for Off-Policy Evaluation, Masatoshi Uehara, Jiawei Huang, and Nan Jiang

---

> > > > > > > ### Comment · Area_Chair_45QX · 2021-09-02
> > > > > > > **Error for w and q?**
> > > > > > >
> > > > > > > I see. So S51y's high-level point is that, with linear functions, if you plug $\hat{w}$ and $\hat{q}$ learned using LSTDQ into the DR form, you still get LSTDQ. So when analyzing LSTDQ, one can invoke the DR analysis and show that the error is the multiplication of error in w and error in q. However, are there existing expressions of $\epsilon\_{\hat w}$ and $\epsilon\_{\hat q}$ that one can plug into $\mathbb{E}\_{d\_0}[\epsilon_{\hat w} \epsilon_{\hat q}]$?
> > > > > > >
> > > > > > > The authors are also welcome to respond to this comment.

---

> > > > > > > > ### Comment · Reviewer_S51y · 2021-09-02
> > > > > > > > **Error for w and q**
> > > > > > > >
> > > > > > > > That is a good point. I'm feeling that one can directly use the definition of error in Theorem 3.1 in [1] to define $\epsilon_{\hat w}$ and $\epsilon_{\hat q}$:
> > > > > > > > $$
> > > > > > > > \epsilon_{\hat w}(s,a) := \frac{d_\pi(s,a)}{d_{\pi_e}(s,a)}-\hat w(s,a),\quad
> > > > > > > > \epsilon_{\hat q}(s,a) := \hat q(s,a)-r-\gamma E[\hat q(s',\pi_e)]
> > > > > > > > $$
> > > > > > > > where $\hat w$ and $\hat q$ are obtained from the derivation in my previous response.
> > > > > > > >
> > > > > > > >
> > > > > > > > However, given that $\hat w$ and $\hat q$ in my previous response are obtained by minimizing two quadratic functions according to [2], which are different from the loss functions in Def 8 & 9 in this paper, it's unclear to me whether the biases in the definition above and Def 8 & 9 in the paper are comparable and whether the difference between them is non-trivial or not (e.g. Def 8 & 9 provides a tighter bound than the above definition). It would be great if the authors can provide some insights on this issue.

---

> > > > > > > > > ### Author Response · Authors · 2021-09-02
> > > > > > > > > **Nontriviality**
> > > > > > > > >
> > > > > > > > > Thank you very much for the detailed explanation.
> > > > > > > > > We checked the derivation and we are convinced that it is correct (at least in essense).
> > > > > > > > >
> > > > > > > > > As for the comparison to Theorem 6, they must coincide with each other
> > > > > > > > > if both $\epsilon_{\hat{w}}(s,a)$ and $\epsilon_{\hat{q}}(s, a)$ are correctly asymptotically evaluated.
> > > > > > > > > In particular, it is not affected by how $\hat{w}$ and $\hat{q}$ are obtained.
> > > > > > > > > This is because Theorem 6 gives the *exact* asymptotic bias of LSTDQ, not an upper bound,
> > > > > > > > > whereas $v_{DR}$ is just proved (by you) to be equivalent to LSTDQ and Theorem 3.1 in [1] gives identity, rather than inequality.
> > > > > > > > > Accordingly, if the evaluation of $\epsilon_{\hat{w}}(s,a)$ and $\epsilon_{\hat{q}}(s, a)$
> > > > > > > > > is done with inequalities, the resulting bound is necessarily no more accurate than Theorem 6 in asymptotic sense.
> > > > > > > > >
> > > > > > > > > Note that the evaluation of $\epsilon_{\hat{w}}(s,a)$ and $\epsilon_{\hat{q}}(s, a)$ is not quite trivial under unrealizability.
> > > > > > > > > If the density-ratio/value-function models contains the true ones $w^*$ and $q^*$, then it is straightforwardly seen
> > > > > > > > > that $\hat{w}$ and $\hat{q}$ asymptotically coincides with them (i.e., $\hat{w}\to w^*$ and $\hat{q}\to q^*$),
> > > > > > > > > which are identical to the "least squares solutions" $f^*$ in Definition 8 and 9.
> > > > > > > > > On the contrary, if the models are misspecified (i.e., $w^*$ and $q^*$ do not exist), there is no trivial guarantee that $\hat{w}$ and $\hat{q}$ have such "least-squares" nature.
> > > > > > > > >
> > > > > > > > > Therefore, we believe the suggested derivation of the double robustness of LSTDQ
> > > > > > > > > gives another, yet nontrivial proof strategy to get to the same conclusion.
> > > > > > > > > Since it is nontrivial, we believe the significance and novelty of Theorem 6 still stand.

---

> > > > > > > > > > ### Comment · Area_Chair_45QX · 2021-09-05
> > > > > > > > > > **BQ# - q**
> > > > > > > > > >
> > > > > > > > > > I would appreciate it if the authors can respond to the following question if they are still able to post new comments (and since the response period has officially concluded, I will not count it against the paper if you do not have a definitive answer to the question). Unlike Def 8 which is clearly the realizability error of density ratio, Def 9 is a little bit hard to parse: it is unclear/unintuitive why the residual $\mathcal{R}_B$ is 0 under value-function realizability (though I think $Q^{\\#}=Q^\pi$ when $Q^\pi$ is realizable and therefore the residual will be $0$); the reference to $Q^{\\#}$ in the definition is somewhat undesirable since the quantity is a complicated expression in the estimator and does not make direct sense in the MDP. Is it possible to rewrite it as an expression that looks like more standard forms of Bellman errors, e.g., replacing $B Q^{\\#} - q$ with $B q - q$ (up to some modifications)?

---

> > > > > > > > > > > ### Author Response · Authors · 2021-09-05
> > > > > > > > > > > **On BQ# - q**
> > > > > > > > > > >
> > > > > > > > > > > Thank you for the question.
> > > > > > > > > > > We are not sure if it is ok to replace $Q^\sharp$ (in $BQ^\sharp-q$) with $q$,
> > > > > > > > > > > which we believe implies that the *infinite-horizon-only* realizability is sufficient for $\mathcal{R}\_B=0$.
> > > > > > > > > > > In fact, what we can currently show is that the *all-horizon* realizability is a sufficient condition,
> > > > > > > > > > > where the value functions truncated by time horizon $H$ is realizable for all $H\ge 0$.
> > > > > > > > > > > Note, however, that this condition is essentially the same as Assumption 1 in [1]
> > > > > > > > > > > and thus the comparison to [1] is still meaningful.
> > > > > > > > > > >
> > > > > > > > > > > See below for the proof.
> > > > > > > > > > >
> > > > > > > > > > > ----
> > > > > > > > > > >
> > > > > > > > > > > Let $Q\_H(s,a):=\sum\_{h=0}^{H-1} \gamma^h (P^h\bar{r})(s,a)$ denote the true value function of $\pi$ up to time horizon $H\ge 0$.
> > > > > > > > > > > We assume that $Q\_H$ is realizable for all $H\ge 0$, i.e., $Q\_H\in \mathcal{F}\_\phi$.
> > > > > > > > > > >
> > > > > > > > > > > We can also think of the finite-horizon version of $Q^\sharp$,
> > > > > > > > > > > $Q^\sharp\_H(s,a):= \sum\_{h=0}^{H-1}\gamma^h b^{\sharp\top}F^{\sharp h}\phi(s, a)$.
> > > > > > > > > > > Then, we can show $[Q^\sharp\_H]\_{H\ge 0}$ is the solution to
> > > > > > > > > > > the oracle FQE formula,
> > > > > > > > > > > $$ Q^\sharp\_{H+1}=\arg \min\_{q\in\mathcal{F}\_\phi} |BQ^\sharp\_H-q|^2. \qquad (1)$$
> > > > > > > > > > > Here we omit the proof of this statement since
> > > > > > > > > > > it is proved with the naive extension of Proposition 5 (from the finite sample average to the true expectation).
> > > > > > > > > > >
> > > > > > > > > > > Next, we show $Q^\sharp\_H=Q\_H$ for all $H\ge 0$.
> > > > > > > > > > > This is done by induction in the FQE fashion.
> > > > > > > > > > >
> > > > > > > > > > > **Case of** $H=0$)
> > > > > > > > > > > Clearly, $Q^\sharp\_0=Q\_0=0$.
> > > > > > > > > > >
> > > > > > > > > > > **Case of** $H\to H+1$)
> > > > > > > > > > > Note that
> > > > > > > > > > > $Q\_{H+1}=\arg \min\_{q\in\mathcal{F}\_\phi} |BQ\_H-q|^2$
> > > > > > > > > > > by the realizability assumption.
> > > > > > > > > > > Then, the assumption of induction implies
> > > > > > > > > > > $Q\_{H+1}=\arg \min\_{q\in\mathcal{F}\_\phi} |BQ^\sharp\_H-q|^2$.
> > > > > > > > > > > On the other hand, we have already seen from $(1)$ that $Q^\sharp\_{H+1}$ also attains the same minumum,
> > > > > > > > > > > which implies $Q^\sharp\_{H+1}=Q\_{H+1}$.
> > > > > > > > > > >
> > > > > > > > > > > Finally, Since $Q\_\infty := \lim\_{H\to\infty}Q\_H$ always exists (note that reward is bounded),
> > > > > > > > > > > the limit of $Q^\sharp\_H$ also exists and coincides with $Q^\sharp$ by definition.
> > > > > > > > > > > This implies
> > > > > > > > > > > $ Q^\sharp(s,a)=\sum\_{h=0}^\infty \gamma^h (P^h\bar{r})(s,a)$,
> > > > > > > > > > > and therefore $BQ^\sharp=Q^\sharp\in\mathcal{F}\_\phi$.
> > > > > > > > > > > This shows $BQ^\sharp-q=Q^\sharp-q=0$ (we used the fact $q$ (or $f^*$ in Def 9) is the projection).
> > > > > > > > > > >
> > > > > > > > > > >
> > > > > > > > > > > QED
> > > > > > > > > > >
> > > > > > > > > > > -----
> > > > > > > > > > >
> > > > > > > > > > >
> > > > > > > > > > > [1] Wang, R., Foster, D., and Kakade, S. M. (2021). What are the statistical limits of offline RL with
> > > > > > > > > > > linear function approximation? In International Conference on Learning Representations.

---

> > > > > > > > > > > > ### Comment · Area_Chair_45QX · 2021-09-05
> > > > > > > > > > > > **Thanks**
> > > > > > > > > > > >
> > > > > > > > > > > > Thanks for the quick reply, and it will be good to include the proof in the paper.
> > > > > > > > > > > >
> > > > > > > > > > > > I am not sure [1] applies here because you additionally assume the invertibility of $I - \gamma F^{\\#}$, which [1] probably doesn't have? Also, when $Q^\pi \in \mathcal{F}_\phi$ and with the invertibility condition, I think you will have $Q^{\\#} = Q^\pi$ (and thus the residual is 0), because LSTDQ can be viewed as solving a linear equation $\Sigma (I - \gamma F^{\\#}) \theta = \mathbb{E}[\phi(s,a)\cdot r]$. When $\theta$ is the true linear coefficients for $Q^\pi$ the equation is obviously satisfied. With invertibility, the solution must be unique (and is therefore $Q^\pi$). For reference, in LSTD (on-policy) the learned function (the counterpart of your $Q^{\\#}$) converges to $V^\pi$ in the realizable case (http://chercheurs.lille.inria.fr/~munos/papers/files/lstd-icml2010.pdf).
> > > > > > > > > > > >
> > > > > > > > > > > > Also, another motivation for avoiding $Q^{\\#}$ in the bias (besides interpretability) is that, if I am allowed to use $Q^{\\#}$, I can always write the bias trivially as $E_{p_0^\pi}[Q^{\\#}] - E_{p_0^\pi}[Q^\pi]$, which is vacuous. In general I don't mean specifically the form $B q - q$ (which I don't think should be correct), but just to point to a direction, e.g., $q - Q^\pi$, or even the error in the "all-horizon" realizability is ok if that turns out to be really needed. I understand this issue may not be solved within a short period of time but I encourage the authors to investigate the issue.

---

> > > > > > > > > > > > > ### Comment · Area_Chair_45QX · 2021-09-05
> > > > > > > > > > > > > **update**
> > > > > > > > > > > > >
> > > > > > > > > > > > > Some updates on the previous comment. My expression may not be accurate b/c the papers' notations are different from what I am used to, but I think the point stands: you let $\theta =$ [the coefficients of] $Q^{\\#}$ (i.e., drop $\phi$), move the inverted matrix to the other side, and verify that the $\theta = $ [coefficients of] $Q^\pi$ is a valid solution. Due to invertibility that will also be the only solution.

---

> > > > > > > > > > > > > > ### Author Response · Authors · 2021-09-05
> > > > > > > > > > > > > > **I see**
> > > > > > > > > > > > > >
> > > > > > > > > > > > > > I see your point. Now I think we can show $Q^\\pi\\in\\mathcal{F}_\\phi$ is sufficient, and the proof will be greatly shorten.

---

> > > > > > > > > > > > > > > ### Comment · Area_Chair_45QX · 2021-09-05
> > > > > > > > > > > > > > > **Great**
> > > > > > > > > > > > > > >
> > > > > > > > > > > > > > > Ok I think we are on the same page then. The authors are still encouraged to explore alternative expressions of the residual that does not refer to $Q^{\\#}$.
> > > > > > > > > > > > > > >
> > > > > > > > > > > > > > > P.S. The lower bound in [1] considers the finite-horizon setting but your setting is discounted. There is a variant of [1] in that setting where the construction is much simpler and easier to reason about: https://arxiv.org/pdf/2011.01075.pdf

---

> > > > > > > > > > > > > > > > ### Author Response · Authors · 2021-09-05
> > > > > > > > > > > > > > > > **Thank You**
> > > > > > > > > > > > > > > >
> > > > > > > > > > > > > > > > > alternative expressions of the residual that does not refer to $Q^\\sharp$ .
> > > > > > > > > > > > > > > >
> > > > > > > > > > > > > > > > Yes, as you said, it may be not possible to solve the issue immediately, but at least we will mention it.
> > > > > > > > > > > > > > > >
> > > > > > > > > > > > > > > > Also, thank you so much for the additional reference.

---

> > > > > > > > > > > > > ### Author Response · Authors · 2021-09-05
> > > > > > > > > > > > > **Thank You**
> > > > > > > > > > > > >
> > > > > > > > > > > > > > it will be good to include the proof in the paper.
> > > > > > > > > > > > >
> > > > > > > > > > > > > Yes, we will include it.
> > > > > > > > > > > > >
> > > > > > > > > > > > > > I am not sure [1] applies here because you additionally assume the invertibility of , which [1] probably doesn't have?
> > > > > > > > > > > > >
> > > > > > > > > > > > > Right, that is why we believe the invertibility of $I-\\gamma F^\\sharp$ is essential to avoid hard instances.
> > > > > > > > > > > > >
> > > > > > > > > > > > > > For reference, in LSTD (on-policy) the learned function (the counterpart of your ) converges to
> > > > > > > > > > > > >  in the realizable case (http://chercheurs.lille.inria.fr/~munos/papers/files/lstd-icml2010.pdf).
> > > > > > > > > > > > >
> > > > > > > > > > > > > > Also, another motivation for avoiding $Q^{\\#}$ in the bias (besides interpretability) is that, if I am allowed to use $Q^{\\#}$, I can always write the bias trivially as $E_{p_0}[Q^{\\#}] - E_{p_0}[Q^\pi]$, which is vacuous. In general I don't mean specifically the form $B q - q$ (which I don't think should be correct), but just to point to a direction, e.g., $q - Q^\pi$, or even the error in the "all-horizon" realizability is ok if that turns out to be really needed. I understand this issue may not be solved within a short period of time but I encourage the authors to investigate the issue.
> > > > > > > > > > > > >
> > > > > > > > > > > > > Thank you for the reference, and so much discussion and valuable suggestions. It will be quite interesting to study if $Q^\pi\in\mathcal{F}\_\\phi$ is sufficient, and we will add the topic to future work.

---

### Official Review · Reviewer_xAW2 · 2021-07-16

**Rating:** 6
**Confidence:** 3

**Summary:**

This paper characterizes the offline policy evaluation (OPE) error with linear function approximation in the discounted reward setting. With certain assumptions, this paper proves that the error of Linear Direct OPE algorithm (which is equivalent to the limit of Fitted Q-Evaluation algorithm) converges to the expectation of the product of two residuals. As a corollary, when either the function class is closed or the density ratio is realizable, the asymptotic error is zero. In addition, in the same setting the paper upper bounds the OPE error for tile-coding estimator.



**Limitations And Societal Impact:**

The authors adequately addressed limitations and potential negative societal impact.

**Main Review:**

This paper proves a novel asymptotic characterization of the OPE error with non-realizable linear function approximation. Compared with previous results, this paper exactly computes the dominant term of the OPE error, instead of providing an upper bound.

On the plus side, this paper is well written. Proofs are technically sound and solid. The results imply that the direct OPE (as well as the Fitted Q-Evaluation) algorithm automatically satisfies doubly-robust-type error bound: when either the Bellman residual is small or the density ratio can be well approximated, the estimation error is small. So this paper could contribute to the understanding of linear OPE tasks.

The main weakness of this paper is that the results are asymptotic (i.e., only holds when the number of samples approaches infinite). It is unclear how realistic are the phenomena described in this paper. For example, the convergence rate in Theorem 6 depends polynomially on the term 1/c_\mu, which is the minimum data density over all state-action pairs. For MDPs with large state/action space, the term could be much larger than the number of samples. In other words, it’s unclear whether the dominating term is still characterized by Eq. (6) in real-world OPE tasks.

In addition, this paper doesn’t directly tackle the difficulty of linear OPE tasks. On the one hand, the assumptions such as 1/c_\mu-dependence or upper bound of \|F^#_\gamma\|_2 exclude some natural hard instances. Are these assumptions an artifact of the analysis, or necessary for the result? On the other hand, the sample complexity of tile-coding estimators is exponential in the dimension d, which is not very realistic.

Minor issues which may cause unnecessary confusion:
-	The notation f* in Definition 8 and 9 is used to denote different functions.
-	Does Definition 3 depend on the policy \pi being evaluated (because the term (P\phi)(s,a) depends on \pi)?


**Time Spent Reviewing:**

3

---

> ### Author Response · Authors · 2021-08-06
> **On Necessity of Boundedness of $F^\sharp_\gamma$**
>
> We thank Reviewer xAW2 for the valuable comments and suggestions.
>
> > On the one hand, the assumptions such as 1/c_\mu-dependence or upper bound of |F^#_\gamma|_2 exclude some natural hard instances. Are these assumptions an artifact of the analysis, or necessary for the result?
>
> A previous study [1] (Theorem 4.1) on the hardness shows that "sufficient exploration (i.e., bounded $1/c_\mu$) + Q-function realizability (i.e., $R_B = 0$)" is not sufficient for learnability of DM. On the contrary, Theorem 6 shows that "bounded $F^\sharp_\gamma$ + sufficient exploration + (either $R_B=0$ or $R_\chi=0$)" is sufficient for learnability of DM. Hence the boundedness of $F^\sharp_\gamma$ is possibly necessary for making DM learnable, although there are the problem of controllability left open.
>
> > Minor issues which may cause unnecessary confusion:
> > - The notation f* in Definition 8 and 9 is used to denote different functions.
> > - Does Definition 3 depend on the policy \pi being evaluated (because the term (P\phi)(s,a) depends on \pi)?
>
> Thank you for pointing out. We will fix the notation accordingly (Definition 3 is dependent on \pi).
>
> **Reference**
>
> [1] Wang, R., Foster, D., and Kakade, S. M. (2021). What are the statistical limits of offline RL with
> linear function approximation? In International Conference on Learning Representations.

---

### Official Review · Reviewer_pMsS · 2021-07-17

**Rating:** 6
**Confidence:** 2

**Summary:**

This is a theoretical paper which addresses the RL off-policy evaluation with linear function approximation. It proves convergence results of linear function approximations and shows that the direct method (model the dynamics) is asymptotically equivalent to FQE (modeling the value function) and analyzes its convergence properties. Additionally, it proves consistency results for such methods under tile-coding, allowing for consistency (using non-parametric estimation) even under unrealizability of the linear approximators.

**Limitations And Societal Impact:**

The authors discuss the limitations of their work.

**Main Review:**

The paper presents interesting results and is well written. I did not check all the proofs in the paper (it is more theoretically oriented than my expertise), and therefore my confidence in the review is limited. It has two major drawbacks - The first and most obvious one is its reliance on linearity. It is uncommon for linear models to properly model MDPs of interest, and therefore I am not sure how impactful the results will be. The authors do provide consistency results using feature tiling which allows for consistency using tile-coding, and therefore the results could in essence be applied to highly nonlinear MDPs, but I expect such case to be limited to low dimensional MDPs in the same way non-parametric always suffer in high dimensions. Second, the set of assumptions the authors use, while not very strong, is very uncommon compared to the majority of OPE work, further limiting the potential of these results to have a large impact on the field.

The central result (theorem 6) is very reminiscent of the properties of the (very commonly used in OPE) doubly robust estimator. The DR is unbiased if either the behavior policy is know (analogous to R_X being realizable), or the control variate is accurate (analogous to R_B being realizable). I am not sure how useful the analogy is in this case, but since it is a very obvious analogy and DR is such a common estimator in OPE it might be worth exploring.

**Time Spent Reviewing:**

~4 hours

---

> ### Author Response · Authors · 2021-08-06
> **Thank You for Comments and Suggestions**
>
> We thank Reviewer pMsS for the valuable comments and suggestions.
>
> > The first and most obvious one is its reliance on linearity. It is uncommon for linear models to properly model MDPs of interest, and therefore I am not sure how impactful the results will be.
>
> It is true linear models are not very general.
> However, note that recent studies on neural tangent kernel (NTK) [1] shows that deep neural networks with large widths can be characterized as linear models. Thus, our result (Theorem 6) is also applicable to FQE with such neural networks by choosing $\mathcal{F_\phi}$ to be NTK.
>
> > Second, the set of assumptions the authors use, while not very strong, is very uncommon compared to the majority of OPE work, further limiting the potential of these results to have a large impact on the field.
>
> We believe Assumption 1-3 are quite natural and common (or weaker than common ones) in the literature, whereas Assumption 4-5 are seemingly not.
> In fact, both of Assumption 4-5 can be dropped if we replace inverses to pseudo-inverses, except they are essential in Proposition 7, which is useful for controlling $F^\sharp_\gamma$. We will update the text to emphasize the difference in their essentiality.
>
> > I am not sure how useful the analogy is in this case, but since it is a very obvious analogy and DR is such a common estimator in OPE it might be worth exploring.
>
> We agree that the connection to DR is worth exploring and the discussion has to be made in our paper.
> Please refer to the separate post on this point (https://openreview.net/forum?id=_y2G1-i7L8&noteId=VPJ6VKy-7z).
>
> **Reference**:
>
> [1] Jacot, Arthur, Franck Gabriel, and Clément Hongler. "Neural tangent kernel: Convergence and generalization in neural networks." arXiv preprint arXiv:1806.07572 (2018).

---

### Official Review · Reviewer_rVKG · 2021-07-20

**Rating:** 4
**Confidence:** 4

**Summary:**

The paper consider off-policy evaluation using linear function approximation
under unrealizability setting.
Under such more relaxed assumption, they provide characterization of the linear direct method(DM)
as an error governed by two approximation residual $\mathcal{R}_B$ and $\mathcal{R}_X$,
$$
    \hat{J}(\pi) - J(\pi) = \mathbb{E}[\mathcal{R}_B(s,a)\mathcal{R}_X(s,a)] + O(1/\sqrt{n}),
$$
with $\mathcal{R}_B$ corresponding to Bellman residual and
$\mathcal{R}_X$ corresponding to residual of marginal density ratio.
Leveraing this result, they establish a nonparametric tile-coding estimators
to solve the unrealizability though with a slower convergence.

**Limitations And Societal Impact:**

yes

**Main Review:**

**Novelty:**
I have the major concern about the novelty of this paper,
where there is a large class of OPE literature about doubly robust estimator, e.g. [1-2],
the paper fails to discuss and compare with.
Especially in [2] Theorem 3.1, the bias term of double robustness estimator can be characterized
as the inner product of two residual terms of Bellman residual and residual of marginal density ratio.
Although the two estimators compared is not exactly the same (doubly robust vs DM),
the bias error decomposition looks quite similar.

I encourage the author to investigate deeper into those papers and find out the relationship.
It is possible that the similar error decomposition can imply linear DM is
already a doubly robust estimator in some sense, which may be a very interesting finding.

**Quality, Clarity and Significance:**
I am satisfied with the quality and the clarity of the paper.
The theoretical significance is still lacking before comparison with doubly robust estimator.

Reference:

[1] Kallus, Nathan, and Masatoshi Uehara. "Double Reinforcement Learning for Efficient Off-Policy Evaluation in Markov Decision Processes."

[2] Tang, Z., Feng, Y., Li, L., Zhou, D., Liu, Q. (2019). Doubly robust bias reduction in infinite horizon off-policy estimation.

**Time Spent Reviewing:**

2

---

> ### Author Response · Authors · 2021-08-06
> **On Connections to DR Estimators**
>
> We thank Reviewer rVKG for the valuable comments and suggestions.
>
> > I encourage the author to investigate deeper into those papers and find out the relationship. It is possible that the similar error decomposition can imply linear DM is already a doubly robust estimator in some sense, which may be a very interesting finding.
>
> We agree the DR literature, specifically both of [1-2], should be discussed in the paper.
> After reading the references, as Reviewer rVKG have speculated, we confirm our Theorem 6 indicates that linear DM is doubly robust in bias on its own. Please see the separate post (https://openreview.net/forum?id=_y2G1-i7L8&noteId=VPJ6VKy-7z) for more detailed comparison (we decided to separate the post so that the other reviewers can notice).

---

### Official Review · Reviewer_Fdr6 · 2021-08-28

**Rating:** 6
**Confidence:** 3

**Summary:**

They analyze the linear DM method in detail. They prove that the estimator has an error form, which is written as an inner product of two residual functions. One is associated with the misspecification of Q-functions. Another is related to the misspecification of W-functions.


**Main Review:**

 I appreciate the author's detailed analysis. In this paper, if I understand correctly, the main new finding of this paper is if you have realizability of marginal ratios and the existence of  I-\gamma F^{#}, you can show the PAC guarantee.

1; if I understand correctly, the non-singularity (existence) of I-\gamma F^{#} is pretty much strong. If you use minimax-type estimators with linear models like modified RBM in CJ 2019 (this paper is for learning but for the modification of policy evaluation is easy), I guess that you can prove the closedness of Q-function class like TQ \subset Q' as you can write like a^{\top} (I-\gamma F) \phi  \subset b^{\top}\phi (Q and Q' are linear models with different radius ) I think this kind of discussion is essential. The present discussion in Section 3.1.1 really does not discuss this point. ( non-singularity of I-gamma F^{#} versus the closedness of Q-function linear class. )

2: Second, UIJKSX 2021 might already show one of the main findings of this paper. They showed that the realizability of marginal ratios and the closedness of the Bellman operator for marginal ratios are sufficient for PAC guarrantee. I think the non-singularity of I- gamma F^{#} essentially says the closedness of the bellman operator for marginal ratios. Pleasee refer to Lemma 4 in UIJKSX. I am sorry if I am wrong at this point.....( Note their estimators are DM methods when using linear models. The estimator is essentially the same as this paper. )

3: Line 341 " To the best of our knowledge, the present study is the first to report the density-ratio-estimation 342 interpretation of linear direct methods in unrealizable settings"
That sounds not correct. For example, if you see UIJKSX 2021, they show the realizability of marginal ratios and Q-functions are sufficient for PAC guarantee by presenting unrealizable settings first (theorem 10). The error takes the product form like the errors for marginal ratios and the errors for Q-functions. (Though this form is different from (6) in the author's paper, you can get rid of the boundedness of F^{#} in their expansion. )


[CJ ]Information-Theoretic Considerations in Batch Reinforcement Learning
https://arxiv.org/abs/1905.00360
[UIJKSX] Finite Sample Analysis of Minimax Offline Reinforcement Learning: Completeness, Fast Rates and First-Order Efficiency https://arxiv.org/pdf/2102.02981.pdf

-----------------------------------------------------

After thinking more, what I wrote regarding the first point and second point are wrong. I apologize for that. Please ignore them.
Somehow, the mysterious assumption, the existence of (I-gamma F^{#}) helps and produces a nice result.


**Time Spent Reviewing:**

2.5

---

### Author Response · Authors · 2021-08-06
**Clarification on the Connection to the DR Literature**

We thank all the reviewers for the helpful comments and suggestions.

Here we separately post the clarification on the relationship between our result and the doubly robust OPE literature.

As all the reviewers pointed out, our result is closely related to the DR results, e.g., [1, 2].
In particular, Theorem 6 is quite similar to Theorem 3.1 of [2] as both establish the double robustness of the bias of some OPE methods.
The differences are the following:
- Theorem 6 shows the DR property of linear DM, while [1, 2] establish those of generic meta algorithms based on given (nice) value-function and density-ratio estimators.
- Theorem 6 is nontrivial w.r.t. [1, 2], because linear DM does not perform any density-ratio estimation explicitly. Algorithm 1 is consistent if the linear function space $\mathcal{F}_\phi$ is correctly specified *either* as a value-function model or a density-ratio model, while performing only value-function estimation. On the contrary, [1, 2] propose meta algorithms that are consistent if either of given two (value-function or density-ratio) estimators are consistent.
- Other minor differences include the generality of the assumptions on data-collecting processes and the availability of behavior policies.

Summarizing, Theorem 6 shows that linear DM is doubly robust in bias on its own. Interestingly, it implies that double robustness can be achieved without estimating density ratio explicitly.


**References**:

[1] Kallus, Nathan, and Masatoshi Uehara. Double Reinforcement Learning for Efficient Off-Policy Evaluation in Markov Decision Processes. (arxiv 2019, JMLR 2020)

[2] Tang, Z., Feng, Y., Li, L., Zhou, D., Liu, Q. Doubly robust bias reduction in infinite horizon off-policy estimation. (arxiv 2019, ICLR 2020)

---

### Comment · Area_Chair_45QX · 2021-08-28
**Quick clarification questions**

Dear authors,

We are still discussing this paper. As you may have noticed (I am not 100% sure you are able to see the new review), I have requested an additional reviewer to look at the paper. Since they performed the review in a relatively short period of time, they had some misunderstandings which were subsequently clarified by discussions among reviewers. Overall the points raised by the new review are covered by existing reviews so there is no need to respond to it if it is visible to you.

I do have 2 more clarification questions:

1. I want to double check that the main result (theorem 6) holds for any MDP, not just in linear MDPs? If I understand correctly, the introduction of linear MDPs in Sec 2.4 is to motivate the direct estimator, but Theorem 6 should hold when the underlying MDP is not linear in the feature map? It is confusing because Theorem 6 relies on Assumptions 1-4, whereas Assumptions 3 and 4 are in the "linear MDP" section, though they have nothing to do with the linear MDP structure (Def 3).

Moreover, in the literature "linear MDP" usually refers to something stronger (see e.g., Duan et al, Jin et al), that $P(s'|s,a) = \phi(s,a)^\top \psi(s')$, which implies and is related to your definition 3. Linear MDPs implicitly induce very stronger assumptions (e.g., one can linearly predict the Bellman backup of any function), so I want to be cautious and verify that your main result does not rely on the true MDP having such structures.

2. Theorem 6 has $1/c_{\mu}$ in its convergence rate, where $c_{\mu}$ is the minimal probability of the data marginal on any state-action pair. This quantity usually appears in the convergence rate of tabular analysis. Linear function approximation is mostly used when the state space is very large, in which case $c_{\mu}$ is necessarily smaller than $1/|S|$ so the convergence rate will have an undesirable $|S|$ dependence. In the linear analysis, I believe it is more common for the convergence rate to depend on e.g., the minimal eigen value of the sample covariance matrix (or, in the case of methods in the LSTD family, perhaps the least singular value of $(I- \gamma F^\sharp)$). Can authors comment on this issue?

thanks,
AC

---

> ### Author Response · Authors · 2021-08-30
> **Thank You for Questions**
>
> Yes, we can see the new review.
> We thank all the reviewers and the area chair for taking extended amount of time for the reviewing process.
>
> > I want to double check that the main result (theorem 6) holds for any MDP, not just in linear MDPs?
>
> Yes, Theorem 6 holds for any MDP.
> All the assumptions in the "linear MDP" section constrain only $\phi$, not the environment $\mathcal{M}$ nor policy $\pi$.
>
> > Theorem 6 has 1/cμ in its convergence rate, where cμ is the minimal probability of the data marginal on any state-action pair. This quantity usually appears in the convergence rate of tabular analysis. Linear function approximation is mostly used when the state space is very large, in which case cμ is necessarily smaller than 1/|S| so the convergence rate will have an undesirable |S| dependence. In the linear analysis, I believe it is more common for the convergence rate to depend on e.g., the minimal eigen value of the sample covariance matrix (or, in the case of methods in the LSTD family, perhaps the least singular value of (I−γF♯)). Can authors comment on this issue?
>
> The minimal eigenvalue (denoted by $c_*$ in Definition 12, Appendix A) plays an essential role to bound the convergence term.
> In fact,
> the convergence rate depends on $1/c_\mu$ *only* through $1/c_*$.
> However, $c_*$ has simultaneous dependencies on $\mu$ (variable of data) and  $\phi$ (variable of estimator).
> We split the condition $1/c_*<\infty$ into $1/c_\mu<\infty$ (Assumption 2) and the irreducibility (Assumption 4)
> to decouple the assumptions on data and the assumptions on estimator. We believe such distinction is important in unrealizable settings.
>
> Below are more technical details:
> - The exact $c_*$-dependency of the convergence rate is seen in the appendix, specifically in Theorem 11, or more explicitly in Eq. (8) at Line 494.
> - Proposition 10 shows the boundedness of $1/c_*$ under the boundedness of $1/c_\mu$ and irreducibility.
> - More quantitatively, we have $c_* \ge c_\mu \inf_{v} || v^\top\phi ||^2_{L^2(\mathcal{S}\times \mathcal{A})}$ according to the proof, where the infimum w.r.t. $v$ is taken over the unit sphere of $\mathbb{R}^K$. The irreducibility is used to bound $\inf_{v} || v^\top\phi ||^2_{L^2(\mathcal{S}\times \mathcal{A})}$ from below.
>
> ----
>
> Edit: Fixed minor error in the technical details, regarding to $\inf_{v} || v^\top\phi ||^2_{L^2(\mathcal{S}\times \mathcal{A})}$.
>
> Edit2: Eq. (9) -> Eq. (8)

---

> > ### Comment · Area_Chair_45QX · 2021-08-30
> > **Thanks**
> >
> > Thanks for the clarification. Personally I'd think $1/c_*$ is the correct dependence, and "$c_*$ depending on both $\mu$ and $\phi$" is a feature instead of a "bug" and don't require further fixing (e.g. relaxing to $c_{\mu}$), but that's a personal choice. I would recommend though that the authors point out in the main text the existence of this result where the rate only depends on $1/c_*$ instead of $1/c_\mu$.

---

> > > ### Author Response · Authors · 2021-08-30
> > > **Thank You**
> > >
> > > > I would recommend though that the authors point out in the main text the existence of this result where the rate only depends on 1/c∗ instead of 1/cμ.
> > >
> > > Thank you for the feedback. We agree and will update the text accordingly.

---

### Decision · Program_Chairs · 2021-09-27

**Decision:**

Accept (Poster)

**Comment:**

The paper characterizes the asymptotic bias of off-policy evaluation using linear function approximation when realizability assumptions do not hold, with convergence rate analysis. The major concerns of the reviewers focused on the paper's lack of discussions of closely related results in MIS (e.g., Uehara et al, Tang et al); in fact, some of the high-level claims in the paper (e.g., first to discover connection between direct methods and density-ratio learning, first to provide "doubly robust" guarantee to LSTDQ) are already known in the prior works.

After extensive post-rebuttal discussions, the AC decides that this is borderline paper that can go either way, but their personal recommendation is acceptance: the paper's exact characterization of the bias in LSTDQ is a significant result. As mentioned above, prior works have noticed the curious nature of LSTDQ and how it connects to MIS methods, but an exact characterization of its bias is missing, and different approaches to bounding the bias (e.g., under either realizability of value function or density ratio) yield different results, lacking unification.

This paper solves this problem by providing the *exact* characterization of the bias, which takes the form of the product of realizability errors of value function and density ratio, respectively. For reference, the most comparable results that is previously known takes the form of minimum over these errors; see post-rebuttal discussions (also Remark 2 in Appendix A of Uehara et al, which does not require realizability). While multiplicative errors are known for doubly robust estimators (e.g., Tang et al, Kallus and Uehara), these are meta-estimators and require approximate value function and density ratio as inputs, so the results are somewhat orthogonal to the current paper. Furthermore, the paper complements the bias analysis by a convergence rate analysis, which takes care of non i.i.d. data and finite-sample effects. With all these, the AC feels that the paper will be a valuable reference for anyone who wants to investigate the statistical behavior of linear OPE estimators, and therefore recommends acceptance.

Despite the AC's recommendation, the authors are noted that missing significant related works is a major issue, and a rejection will also be reasonable given that the revision needed is quite substantial that may warrant another round of reviewing. Regardless of the final decision, the authors are urged to fix overclaiming and draw connections to related works. Just to give a few pointers:

- Abstract: "OPE ... has been advanced under the (approximate) realizability assumptions. However, such assumptions undermine the applicability of the results since the given environmental models may be completely wrong." This sentence suggests a qualitative difference from existing works, whereas the actual difference is more quantitative.

- Line 60: " new interpretation of the linear DMs as marginal density ratio estimator" Already known in the literature.

- A comparison to previous upper bounds on the bias should be provided.

Finally, some additional detailed comments from the AC:

1. The authors might want to reconsider the title, which is not very informative in its current form given that the central result is the exact characterization of bias and the connection between DM (LSTDQ) and MIS is somewhat known. Also, linear function approximation is a crucial component of the setup, which is not reflected in the title.

2. The authors mentioned the lower bound of Wang et al several times during discussions, but may have some slight misunderstanding of their results. It was mentioned that they assumed bounded $1/C_{\mu}$, which is not true. Wang et al's data assumption is about the minimal eigenvalue of $\Sigma$, which is a much weaker one; the authors should be careful when discussing related issues in the paper. On a somewhat related note, the paper's derivation requires invertible $\Sigma$, but that may not be necessary. In the usual LSTD analyses, $\Sigma$ and $(I-\gamma F^{\\#})$ is often multiplied together as a single matrix (usually in the form of $E[\phi(s)\phi(s) - \gamma \phi(s) \phi(s')]$, and the requirement of invertibility of $\Sigma$ may be an artifact of extracting out $\Sigma$.

3. Def 9 for "Bellman residual" is not as interpretable as Def 8. For Def 8, one can easily see that the residual is 0 if the linear class realizes the true density ratio function. For Def 9, it is unclear/unintuitive why the residual is 0 when the linear class realizes the true value function (though I think $Q^{\\#}=Q^\pi$ when $Q^\pi$ is realizable and therefore the residual is $0$ in this case); the reference to $Q^{\\#}$ in the definition is somewhat undesirable since the quantity is a complicated expression in the estimator which does not make direct sense in the MDP. Also, if $Q^{\\#}$ is allowed in the expression, bias can be trivially written as $E_{p_0^\pi}[Q^{\\#} - Q^\pi]$ which is vacuous. Is it possible to rewrite it as an expression that looks like more standard forms of Bellman errors, e.g., replacing $B Q^{\\#} - q$ with $B q - q$?

Uehara et al. Minimax Weight and Q-Function Learning for Off-Policy Evaluation.

Kallus and Uehara. Double Reinforcement Learning for Efficient Off-Policy Evaluation in Markov Decision Processes.

Tang et al. Doubly robust bias reduction in infinite horizon off-policy estimation.

Wang et al. What are the Statistical Limits of Offline RL with Linear Function Approximation?